# Gene-centric functional dissection of human genetic variation uncovers regulators of hematopoiesis

Satish K Nandakumar[1,2,3†], Sean K McFarland[1,2,3†], Laura M Mateyka[1,2,3,4‡],
Caleb A Lareau[1,2,3,5‡], Jacob C Ulirsch[1,2,3,5‡], Leif S Ludwig[1,2,3],
Gaurav Agarwal[1,2,3,6,7], Jesse M Engreitz[3,8], Bartlomiej Przychodzen[9],
Marie McConkey[10], Glenn S Cowley[3], John G Doench[3], Jaroslaw P Maciejewski[9],
Benjamin L Ebert[3,10,11,12], David E Root[3], Vijay G Sankaran[1,2,3,7]*

[1]Division of Hematology/Oncology, Boston Children's Hospital, Harvard Medical
School, Boston, United States; [2]Department of Pediatric Oncology, Dana-Farber
Cancer Institute, Harvard Medical School, Boston, United States; [3]Broad Institute of
MIT and Harvard, Cambridge, United States; [4]Biochemistry Center (BZH), Ruprecht-
Karls-University Heidelberg, Heidelberg, Germany; [5]Program in Biological and
Medical Sciences, Harvard Medical School, Boston, United States; [6]University of
Oxford, Oxford, United Kingdom; [7]Harvard Stem Cell Institute, Cambridge, United
States; [8]Harvard Society of Fellows, Harvard University, Cambridge, United States;
[9]Department of Translational Hematology and Oncology Research, Taussig Cancer
Institute, Cleveland Clinic, Cleveland, United States; [10]Division of Hematology,
Brigham and Women's Hospital, Boston, United States; [11]Department of Medical
Oncology, Dana-Farber Cancer Institute, Boston, United States; [12]Howard Hughes
Medical Institute, Chevy Chase, United States

*For correspondence:
sankaran@broadinstitute.org

[†]These authors contributed
equally to this work
[‡]These authors also contributed
equally to this work

**Competing interests:** The
authors declare that no
competing interests exist.

**Reviewing editor:** Stephen
Parker, University of Michigan,
United States

**Abstract** Genome-wide association studies (GWAS) have identified thousands of variants
associated with human diseases and traits. However, the majority of GWAS-implicated variants are
in non-coding regions of the genome and require in depth follow-up to identify target genes and
decipher biological mechanisms. Here, rather than focusing on causal variants, we have undertaken
a pooled loss-of-function screen in primary hematopoietic cells to interrogate 389 candidate genes
contained in 75 loci associated with red blood cell traits. Using this approach, we identify 77 genes
at 38 GWAS loci, with most loci harboring 1–2 candidate genes. Importantly, the hit set was
strongly enriched for genes validated through orthogonal genetic approaches. Genes identified by
this approach are enriched in specific and relevant biological pathways, allowing regulators of
human erythropoiesis and modifiers of blood diseases to be defined. More generally, this
functional screen provides a paradigm for gene-centric follow up of GWAS for a variety of human
diseases and traits.
DOI: https://doi.org/10.7554/eLife.44080.001

## Introduction

As genotyping technologies and accompanying analytical capabilities have continued to improve,
genome-wide association studies (GWAS) have identified tens of thousands of variants associated
with numerous human diseases and traits. Despite these advances, our ability to discern the underly-
ing biological mechanisms for the vast majority of such robust associations has remained limited,
with a few exceptions (*Claussnitzer et al., 2015*; *Gupta et al., 2017*; *Mohanan et al., 2018*;

*Musunuru et al., 2010*; *Sankaran et al., 2008*; *Smemo et al., 2014*). In general, published successes have required in-depth mechanistic studies of individual loci and implicated genes to decipher biological mechanisms.

Recent innovations in functional and computational genomics have advanced the field and enabled more rapid and higher-throughput identification of putative causal variants. Approaches that have shown the most success include the use of massively parallel reporter assays to examine allelic variation (*Tewhey et al., 2016*; *Ulirsch et al., 2016*; *Vockley et al., 2015*) and perturbation approaches for dissecting the necessity of regulatory elements (*Canver et al., 2015*; *Chen et al., 2015*; *Fulco et al., 2016*; *Simeonov et al., 2017*). In addition, genetic fine mapping approaches have improved our ability to identify putative causal variants among larger sets of variants in linkage disequilibrium (*Guo et al., 2017*; *International Inflammatory Bowel Disease Genetics Consortium et al., 2017*; *Ulirsch et al., 2019*). However, even when putative causal variants are identified at a disease or trait-associated locus, they most often localize to non-coding regions of the genome, making it difficult to connect variants to genes that mediate the observed effects in a scalable manner (*Claussnitzer et al., 2015*; *Gupta et al., 2017*; *Smemo et al., 2014*).

In the context of hematopoiesis, GWAS studies have identified thousands of variants associated with various blood cell traits, including hundreds associated with red blood cell traits alone (*Astle et al., 2016*; *van der Harst et al., 2012*). Thorough follow-up efforts at individual loci have identified important regulators of hematopoiesis, such as the key regulator of fetal hemoglobin expression, BCL11A (*Basak et al., 2015*; *Liu et al., 2018*; *Sankaran et al., 2008*). However, as in other tissues, the low-throughput with which associated genetic variants can be connected to target genes underlying phenotypes continues to pose a problem for gaining biological insights and clinical actionability in complex traits and diseases.

To accelerate the rate at which genetic variants can be connected to target genes, high-throughput loss-of-function screens involving putative causal genes underlying the genetic associations can be undertaken. This approach is complementary to conventional variant-focused methods and overcomes bottlenecks that can arise during downstream target gene identification. As a proof-of-principle, we connected variants associated with RBC traits to genes regulating erythropoiesis by directly perturbing all candidate genes in primary human hematopoietic stem and progenitor cells (HSPCs) undergoing synchronous differentiation into the erythroid lineage. We demonstrate unique opportunities to rapidly screen for potential candidate gene mediators and identify networks of biological actors underlying trait-associated variation. We additionally illustrate the value of such screens to uncover previously unappreciated regulators of human hematopoiesis that may serve as key disease modifiers.

## Results

### Design and execution of an shRNA screen using blood cell trait GWAS hits to identify genetic actors in erythropoiesis

We applied a gene-centric loss-of-function screening approach to GWAS of RBC traits. We focused on 75 loci associated with RBC traits that were identified by a GWAS performed in up to 135,000 individuals (*van der Harst et al., 2012*) spanning 6 RBC traits (*Figure 1—figure supplement 1A*). Importantly, these 75 loci have been robustly replicated in more recently reported association studies performed on larger cohorts and thus represent important targets for perturbation studies (*Astle et al., 2016*; *Ulirsch et al., 2019*). We endeavored to select candidate genes that could potentially underlie these 75 GWAS signals. To do this, each of the 75 sentinel SNPs was first expanded to a linkage disequilibrium (LD) block including all SNPs in high LD ($r^2$ >0.8, *Figure 1A*, *Figure 1—figure supplement 1B*), then further to the nearest genomic recombination hotspot. Based upon insights from previous expression quantitative trait locus (eQTL) studies (*Montgomery and Dermitzakis, 2011*; *International Inflammatory Bowel Disease Genetics Consortium et al., 2011*; *Veyrieras et al., 2008*), each gene annotated in the genome was expanded to include a wingspan encompassing 110 kb upstream and 40 kb downstream of the transcriptional start and end sites, respectively, to also capture potential functional regulatory elements. This resulted in selection of 389 genes overlapping or in the vicinity of the LD blocks to be tested in the pooled loss-of-function screen. These were distributed at a median of 4 genes per loci (*Figure 1—*

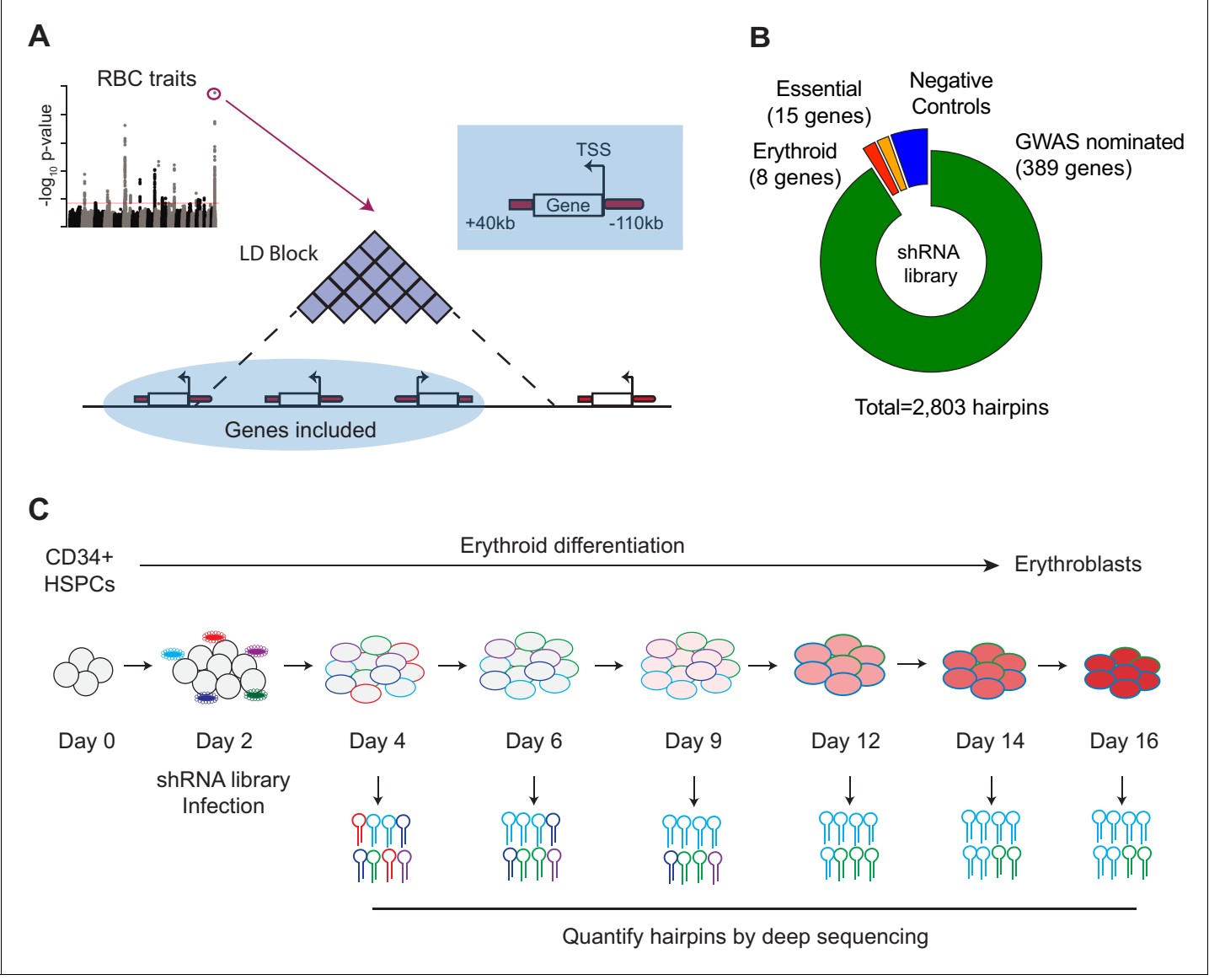

**Figure 1.** Design and Execution of an shRNA Screen Using Blood Cell Trait GWAS Hits to Identify Genetic Actors in Erythropoiesis. (**A**) Overview of shRNA library design.75 loci associated with red blood cell traits (*van der Harst et al., 2012*) were used as the basis to calculate 75 genomic windows of LD 0.8 or greater from the sentinel SNP. Genes with a start site within 110 kb or end site within 40 kb of the LD-defined genomic windows were chosen as candidates to target in the screen. (**B**) Compositional makeup of the library, depicted as number of genes and number of hairpins for each of the four included subcategories; GWAS-nominated genes, erythroid genes, essential genes, and negative control genes (*Figure 1—source data 2*). (**C**) Primary CD34+hematopoietic stem and progenitor cells (HSPCs) isolated from three independent donors were cultured for a period of 16 days in erythroid differentiation conditions. At day 2, cells were infected with the shRNA library, and the abundances of each shRNA were measured at days 4, 6, 9, 12, 14, and 16 using deep sequencing.

DOI: https://doi.org/10.7554/eLife.44080.002

The following source data and figure supplements are available for figure 1:

**Source data 1.** Table containing annotations and information for the 75 SNPs used to seed the shRNA library.

DOI: https://doi.org/10.7554/eLife.44080.006

**Source data 2.** Table containing annotations and information for all hairpins, as well as shRNA counts for each time point and replicate.

DOI: https://doi.org/10.7554/eLife.44080.007

**Figure supplement 1.** Characteristics of GWAS Loci and Gene Selection for Pooled Screen.

DOI: https://doi.org/10.7554/eLife.44080.003

**Figure supplement 2.** Feasibility of Loss of Function Approaches to Perform Pooled Screens in Primary Hematopoietic Stem and Progenitor Cells (HSPCs).

*Figure 1 continued on next page*

*Figure 1 continued*

DOI: https://doi.org/10.7554/eLife.44080.004

**Figure supplement 3.** Pooled shRNA screen in primary HSPCs undergoing erythroid differentiation.

DOI: https://doi.org/10.7554/eLife.44080.005

*figure supplement 1C*). This approach may on occasion miss genes targeted by a trait-associated regulatory element. For example, our approach would miss long-range interactions, as is observed at the *FTO* locus with *IRX3/IRX5* (*Claussnitzer et al., 2015*; *Smemo et al., 2014*). It is becoming increasingly clear that LD and related metrics will only nominate a fraction of potential regulatory targets (*Whalen and Pollard, 2019*). However, there still exists a nontrivial amount of valid targets within reach of proximity LD approaches, especially when the calculation of such windows are extended to reach the nearest recombination hotspot, suggesting that our approach would capture many candidate target genes.

Since the majority of common genetic variation underlying RBC traits appears to act in a cell-intrinsic manner within the erythroid lineage, we decided to perturb the candidate genes during the process of human erythropoiesis (*Giani et al., 2016*; *Sankaran et al., 2012*; *Sankaran et al., 2008*; *Ulirsch et al., 2016*). We chose a pooled short hairpin RNA (shRNA) based loss-of-function approach in primary hematopoietic cells to leverage a number of distinct strengths. First, we have had prior success validating individual genes underlying RBC traits using shRNA-based approaches in primary CD34$^+$ HSPC-derived erythroid cells and our results have been consistent with orthogonal CRISPR based approaches (*Giani et al., 2016*). Second, shRNA libraries can be much more efficiently packaged into lentiviruses and delivered to primary HSPCs compared to alternative CRISPR/Cas9-based guide RNA libraries (*Ting et al., 2018*). We observe that all in one CRISPR or CRISPRi lentiviral constructs that are ideal for primary HSPC screens produce low titer viruses and require very high multiplicities of infection that can be challenging to achieve for a pooled functional screen (*Figure 1—figure supplement 2A,B*). Third, the shRNA approach avoids potential complications like non-uniform loss-of-function or gain-of-function outcomes produced by CRISPR/Cas9 based approaches due to unpredictable DNA repair processes (*Figure 1—figure supplement 2C*) (*Mandegar et al., 2016*). Furthermore, shRNAs can act rapidly to achieve gene knockdown and thereby avoid compensatory effects that can occur when complete CRISPR knockout is achieved (*Rossi et al., 2015*), better recapitulating the subtle changes in gene expression that are characteristic of common genetic variation.

Mobilized peripheral blood-derived primary human CD34$^+$ HSPCs from three independent healthy donors were infected with a lentiviral-based pooled shRNA library consisting of 2803 hairpins targeting the 389 GWAS-nominated genes, along with 30 control genes (*Moffat et al., 2006*). Each GWAS-nominated gene was targeted with 5–7 distinct shRNAs (*Figure 1—figure supplement 3A*). The set of control shRNAs encompassed essential housekeeping genes as positive controls, negative controls in the form of luciferase and other genes not expressed in humans, and a well-established set of genes known to be important for erythropoiesis (erythroid controls) (*Figure 1B*). Using lentiviral libraries with defined titers, we achieved an infectivity of 35–50%, which provides a good tradeoff between reasonably high infection while minimizing the possibility of multiple integrations per cell that can lead to combinatorial phenotypes. To achieve sufficient library representation, we infected at least 1000 CD34$^+$ HSPCs per hairpin (7 ~ 11 * 10$^6$ cells per experiment). The infected HSPCs were cultured using a three-phase semi-synchronous erythroid differentiation method where differentiation blockade reduces cell numbers either through cell death or through a failure of proliferation (*Giani et al., 2016*; *Hu et al., 2013*). We hypothesized that hairpins targeting potential regulators of erythropoiesis would be depleted or enriched during the three-phase erythroid culture, similar to our prior experience in analyzing specific GWAS-nominated genes (*Giani et al., 2016*; *Sankaran et al., 2012*; *Ulirsch et al., 2016*). To assay these hairpins, we isolated and deep-sequenced genomic DNA from the pool of infected cells at six different culture time points that represent distinct stages of erythropoiesis to most broadly assess putative causal genes that may act across the span of differentiation (*Figure 1C*, *Figure 1—figure supplement 3B*).

## Summary characterization of shRNA screen outcomes

For the vast majority of the ~3000 hairpins included in the library, infection was efficient and consistent. Greater than 95% of hairpins were represented at levels of at least five $\log_2$ counts per million (CPM) at day 4, two days post-infection (*Figure 2A*). Across the two-week time course, a diversity of effects - in terms of both increased and decreased hairpin abundance - were observed. While many hairpins were selected against during the course of erythroid differentiation, as reflected in decreases of those hairpin abundances over time, there were also a number of hairpins that increased in the culture over the time course (*Figure 2B*, *Figure 2—figure supplement 1*).

The tested set of hairpins targeting genes nominated by the 75 loci showed a variety of activities, forming a broad distribution spanning both decreases and increases in abundance at different time points (*Figure 2B*). The various controls included in the library behaved as expected. Hairpins targeting genes with known biological roles in erythropoiesis, such as *GATA1* and *RPS19* (*Khajuria et al., 2018*; *Ludwig et al., 2014*), showed markedly decreased abundance across the time course. Likewise, hairpins targeting a set of broadly essential genes (*Figure 1—source data 2*) were strongly depleted by day 16 when compared to negative control hairpins targeting non-human genes, which showed little if any change (*Figure 2C–E*). These trends were recapitulated with strong correlation in each of the three donor CD34$^+$ cell backgrounds (*Figure 2—figure supplement 2*).

## Statistical modeling of gene effects and accounting for confounders in the shRNA screen

The resulting observations of hairpin abundance at each time point were used to model the importance of each targeted gene during the process of erythropoiesis. A linear mixed model was implemented to account for the longitudinal nature of the time course data (*Li et al., 2015*) and to handle the confounding off-target and efficiency effects inherent to the shRNA modality (*Riba et al., 2017*; *Tsherniak et al., 2017*). Since we wanted our model to be able to detect significant changes in hairpin abundance at any time point throughout the differentiation process, we converted the absolute hairpin abundances at each of the six time points to a $\log_2$ fold change relative to the initial hairpin abundances at the start of the differentiation. Using this metric as our response variable, we pooled together the observations for each of the three donor replicates and specified a fixed effect for each gene to capture the contribution that suppressing it with shRNAs would have on the respective abundances for each of the resulting five time intervals. Given the potential variability that could emerge by using shRNAs, we fit a random effect for each hairpin to minimize the chance of conflating inefficiency or off-target effects with the specific on-target gene effect.

After fitting this model to the data, we selected our hit set using a two-threshold approach in which both the magnitude and statistical confidence of the estimated gene effect size were considered. Specifically, genes were called as hits if they had a fitted slope >0.1 $\log_2$ fold change per day within the interval while simultaneously possessing a Wald chi-square FDR-adjusted q value < 0.1. This combined approach allowed us to avoid focusing on genes with large, but highly variable or conflicted effects, as well as genes with highly confident but miniscule effects. In total, this approach identified 77 genes at 38 of the 75 targeted loci which, when suppressed, had a significant effect on the slope of shRNA-encoding DNA abundance at any point during the time course. A majority of these hit loci (27 loci) had 1–2 gene targets prioritized (*Figure 3A*, *Figure 3—figure supplement 1A*). These candidate genes were found to be distributed across all 6 of the originally annotated RBC GWAS traits (*Figure 3—figure supplement 1B*), and hairpins targeting them showed strong internal consistency (*Figure 3—figure supplement 1C*).

To evaluate the validity of this hit set, we began by assaying for enrichment of erythroid essentiality, as recently quantified for each gene in the K562 erythroid cell line (*Wang et al., 2015*). A permutation comparing the sum of K562 essentiality scores for the hit genes with those of randomly drawn, identically-sized gene sets from the library of targeted genes revealed that the hit set was indeed enriched with p=0.0269 (*Figure 3B*). Likewise, when compared to permuted sets of 77 genes randomly chosen from the genome (*Figure 3—figure supplement 2A*), there was even stronger enrichment for erythroid essentiality with p=0.00021, consistent with the idea that genes in the library likely have stronger essentiality due to their genomic proximity to the GWAS hits. We further explored whether the enrichment could be due to an intrinsic bias inherent to GWAS screening itself by permuting sets of genes from libraries nominated by SNPs associated with low-density

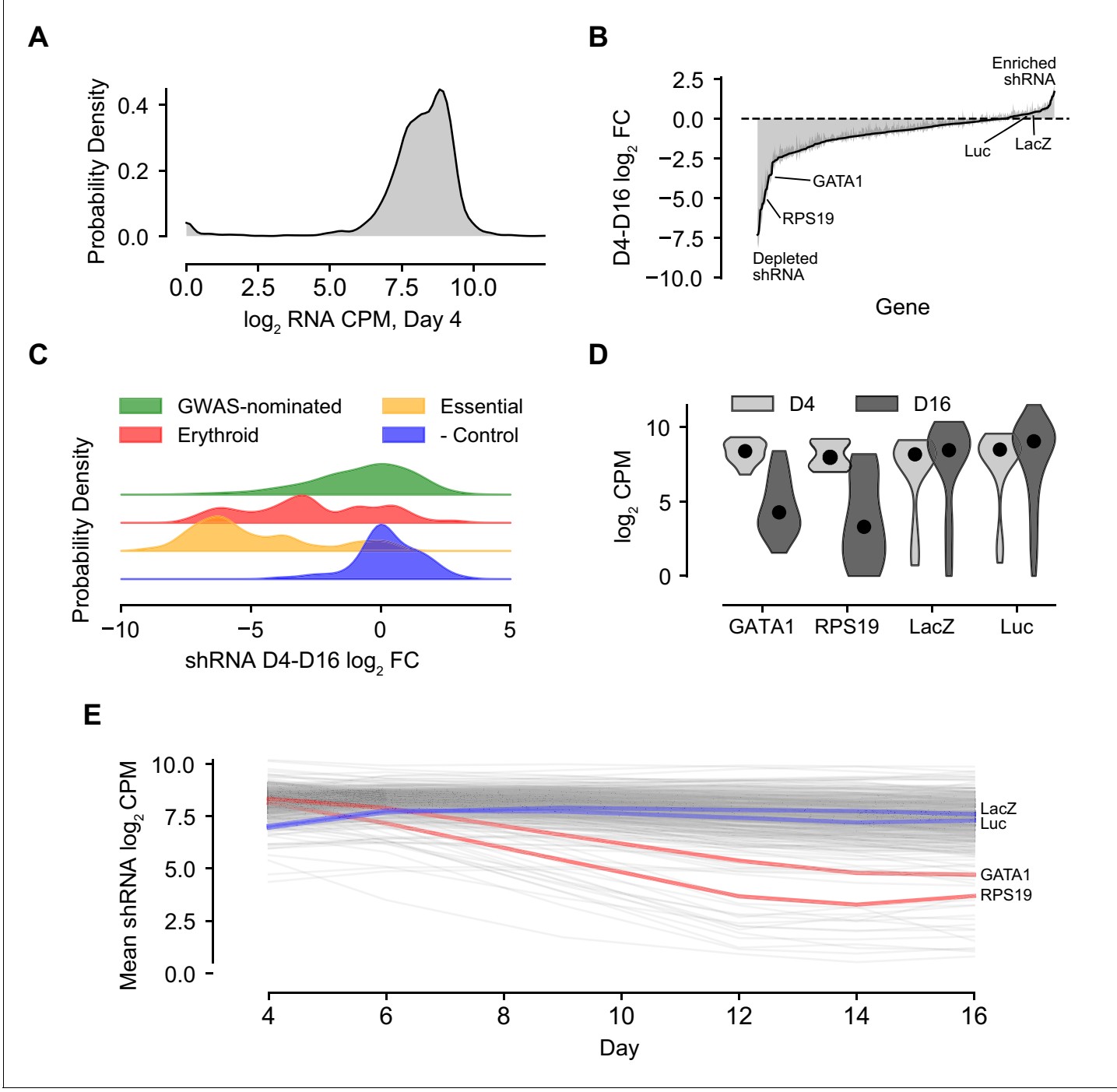

**Figure 2.** Summary Characterization of shRNA Screen Outcomes. (A) Kernel density plot showing library representation as $\log_2$ shRNA CPM across all hairpins. (B) shRNA abundance $\log_2$ fold changes from day 4 to day 16. Represented values are the mean of hairpin abundance $\log_2$ fold changes across hairpins for each gene and two standard deviations. (C) Kernel density plots representing the day 4 to day 16 $\log_2$ fold changes of hairpin abundances for each of the subcategories of the library, including GWAS-nominated genes, known erythroid essential genes, essential genes to cell viability, and orthogonal genes serving as negative controls. (D) Violin plot of day 4 and day 16 $\log_2$ CPM for known actors *GATA1* and *RPS19* and negative controls LacZ and luciferase. (E) $\log_2$ hairpin counts averaged for known actors *GATA1* and *RPS19* as well as negative controls LacZ and luciferase across the course of the experiment. Gray lines depict the universe of all other gene traces in the library for context.
DOI: https://doi.org/10.7554/eLife.44080.008

The following figure supplements are available for figure 2:

**Figure supplement 1.** shRNA abundance $\log_2$ fold changes from day four to each of the other time points.

*Figure 2 continued on next page*

Figure 2 continued

DOI: https://doi.org/10.7554/eLife.44080.009

**Figure supplement 2.** Scatter plots showing agreement of replicate observations across independent CD34⁺ donor populations.

DOI: https://doi.org/10.7554/eLife.44080.010

lipoprotein levels, high-density lipoprotein levels, and triglyceride levels, finding the hit set to be significantly enriched in all comparisons (*Global Lipids Genetics Consortium et al., 2013*) (*Figure 3—figure supplement 2B–D*).

We further validated the ability of this approach to discover genetically relevant hits by performing a permutation analysis based upon five 'gold standard' genes in the library, which possess known genetic underpinnings via identified causal variants: *CCND3* (*Sankaran et al., 2012*; *Ulirsch et al., 2019*), *SH2B3* (*Giani et al., 2016*), *MYB* (*Galarneau et al., 2010*; *Sankaran et al., 2013*; *Sankaran et al., 2011*), *KIT* (*Jing et al., 2008*; *Ulirsch et al., 2019*), and *RBM38* (*Ulirsch et al., 2016*). Calculating the rank sums of hairpins ordered by our model's computed FDR scores for 1,000,000 random combinations of five genes from the library yielded a distribution over which enrichment for the five gold standards was seen with p=0.0249 (*Figure 3C*). While the vast majority of putative causal variants at the RBC trait-associated loci are in non-coding regions, which can be challenging to use to identify a specific target gene, a subset are in coding regions and thereby nominate a specific gene. As a result, we assayed for the presence of coding variants fine-mapped to the interrogated loci from a recent large GWAS that demonstrated a minimum posterior probability of association of 0.1 among the gene hits and compared this with the overall set of genes interrogated in our library (*Ulirsch et al., 2019*). Collectively, these coding variants were found to be 75% missense, 19% synonymous, and 5% frameshift. Among the 389 GWAS-nominated genes in our library, 20 (~5%) were found to contain at least one coding variant from this list. Of these, there was a significant enrichment observed among the hits (~9%, p=0.03907 as determined by permutation analysis; *Figure 3D*).

Having established genetic confidence in our hit set, we next investigated whether the selected genes satisfied enrichment criterion within the erythroid branch of hematopoiesis. RNA expression values for each of the 77 hit genes were examined in datasets spanning human hematopoiesis (*Corces et al., 2016*), as well as adult and fetal erythropoiesis (*Yan et al., 2018*) (*Figure 3E,F*; *Figure 3—figure supplement 3*). For each cellular context, gene expression values were z-score normalized for each gene targeted in the screen. Enrichment was tested through permutation by using the sum of expression z-scores at each stage for the called hits as the benchmark, and comparing these to sums derived from expression values from a matching number of genes randomly drawn from the wider screen. In the more holistic hematopoiesis dataset, common myeloid progenitors (CMPs) and megakaryocyte-erythroid progenitors (MEPs) were significantly enriched for hit gene expression (p<0.01). These progenitor populations are known to contain the progenitors that give rise to erythroid cells. Within a more detailed and separate analysis of human adult erythropoiesis, proerythroblast, early basophilic, and late basophilic erythroblast stages were particularly enriched (p<0.001). The stage at which given genes are implicated to play a role in erythropoiesis from the literature likewise often corresponded with the largest magnitude fold changes across the longitudinal time course measurements, as was the case for earlier genes like *RPL7A*, *RPL23A*, *RPS19*, and *KIT* (*Gazda et al., 2012*; *Jing et al., 2008*; *DBA Group of Société d'Hématologie et d'Immunologie Pédiatrique-SHIP et al., 2012*) as well as late genes like *SLC4A1* and *ANK1* (*Bennett and Stenbuck, 1979*; *Peters et al., 1996*). To examine how our results compare to target gene identification through eQTL-based approaches, we also examined the whole blood eQTL dataset form the Genotype-Tissue Expression (GTEx) Project, finding that none of the 77 shRNA screen hits emerged using eQTLs located within the LD blocks of the original 75 sentinel SNPs (*GTEx Consortium, 2015*). This is not entirely surprising given that the shRNA screen was performed on differentiating erythroid progenitors which are essentially not present in whole blood, so one would expect to miss cell type-specific effects or eQTLs that act in early progenitor populations. Taken together, these results show that this functional gene-centric screen can identify putative causal genes underlying RBC-trait GWAS hits orthogonal to those that would be found with more conventional eQTL-based methods, and which demonstrate clear enrichment in independent genetic and cell biological datasets. We are therefore able to validate the utility of such an approach to identify biologically-relevant genes

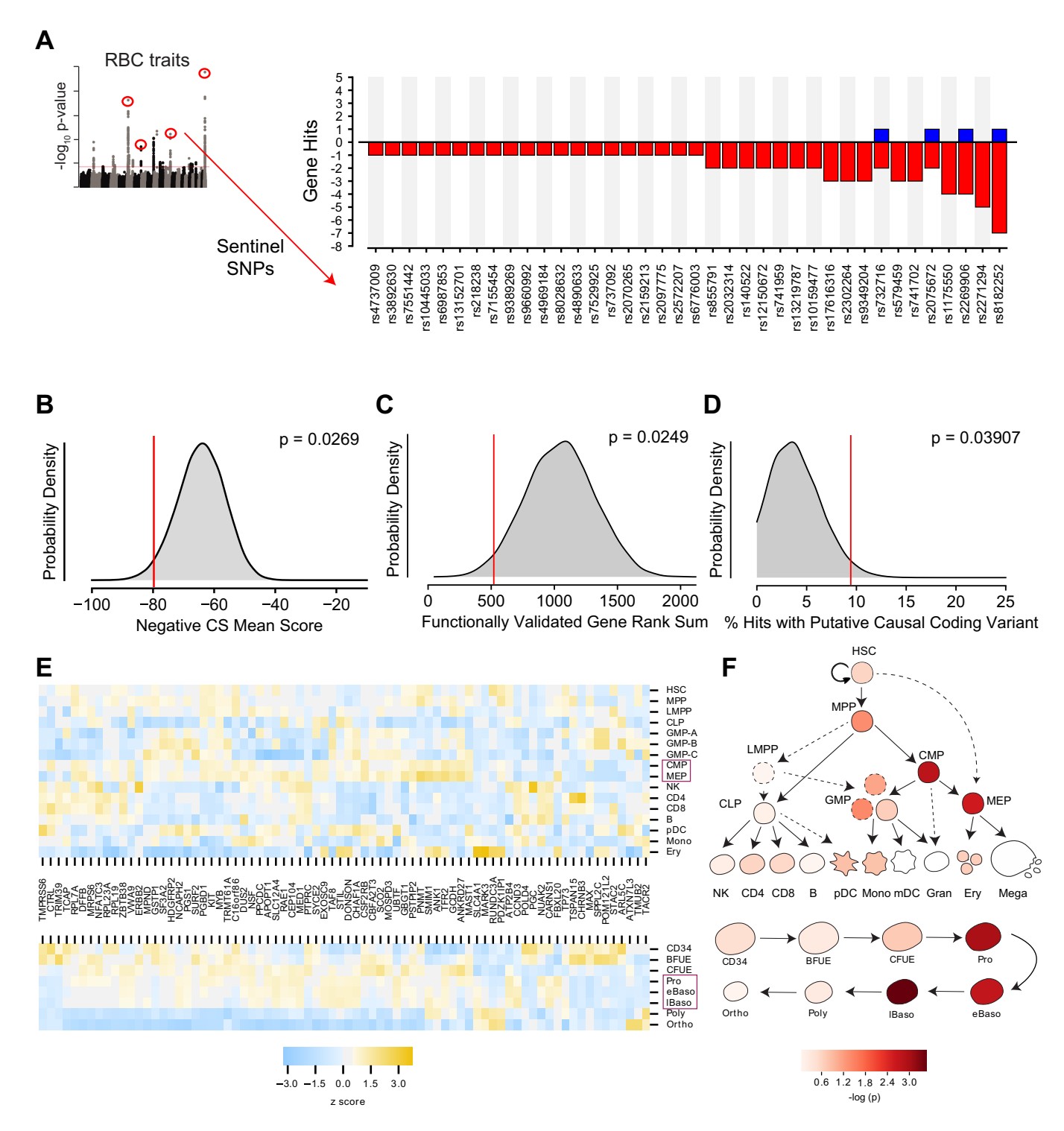

**Figure 3.** Statistical Modeling of Gene Effect Accounting for Off-target shRNA Confounders. (A) Bar graph showing the 38 of 75 loci in the screen with at least one corresponding statistically significant (FDR < 0.1, β >0.1) gene effect causing either a positive or negative log₂ fold change in shRNA abundance.Statistical model output for each gene in screen available in *Figure 3—source data 1*. (B) Kernel density plot showing the expected distributions of K562 essentiality scores using permuted gene hit sets from the library. (C) Hairpin rank sums for permuted sets of 5 genes. The red line indicates the enriched rank sums for 5 'gold standard' genes included in the library, *CCND3*, *SH2B3*, *MYB*, *KIT*, and *RBM38*, for each which a genetic basis of action has already been established. (D) Permuted distribution of % inclusion of predicted coding variants among the set of identified hits. (E) Heat map depicting strength of expression (as z scores within each gene) for each of the 77 identified hit genes across hematopoietic lineages (top)

*Figure 3 continued on next page*

*Figure 3 continued*

and throughout the specific stages of adult erythropoiesis (bottom). Purple boxes highlight the cell types that were enriched for expression of hit genes. (F) Calculated enrichment of the identified hit genes for expression across hematopoietic lineages (top) and throughout the specific stages of adult erythropoiesis (bottom). In both cases, cellular states corresponding to those along the erythropoietic lineage had elevated probability of expressing genes from the hit set as compared to other genes from the library.

DOI: https://doi.org/10.7554/eLife.44080.011

The following source data and figure supplements are available for figure 3:

**Source data 1.** Table containing the R model output for each gene.
DOI: https://doi.org/10.7554/eLife.44080.015

**Figure supplement 1.** Additional Characterization of Modeling Outcomes.
DOI: https://doi.org/10.7554/eLife.44080.012

**Figure supplement 2.** K562 Essentiality Scores Comparing Hit Genes vs.Genes Implicated by Other Traits.
DOI: https://doi.org/10.7554/eLife.44080.013

**Figure supplement 3.** Heat map depicting strength of expression (as z scores within each gene) for each of the 77 identified hit genes throughout the specific stages of fetal erythropoiesis.
DOI: https://doi.org/10.7554/eLife.44080.014

underlying human genetic variation and holistically identify potential stages at which such target genes may act to impact the process of hematopoiesis.

## Analysis of interactions among members of the hit set identifies signaling, structural, and translation-related subnetworks important for erythropoiesis

By screening all loci and genes at once, our approach afforded us the immediate value of examining associations between hits in a holistic fashion, unearthing both familiar and more novel gene cassettes that play a role in erythropoiesis (*Boyle et al., 2017*). Using STRING interaction network analyses (version 10.5) (*Szklarczyk et al., 2017*), we used empirical, database-curated, co-expression, genomic proximity, and text-mined evidence to identify underlying networks between hits in the screen. These networks highlighted a number of interacting biological processes of both known and previously unappreciated importance to erythropoiesis (*Figure 4*), including cell signaling and transcription, cytoskeletal and membrane structure and function, and mRNA translation. We observed a number of molecules that play roles in cell signaling or transcriptional regulation. MYB is a master regulator transcription factor that has been implicated in playing a role in fetal hemoglobin regulation and in erythropoiesis more generally (*Mucenski et al., 1991*; *Wang et al., 2018*). The *MYB* locus has been associated with numerous red blood cell traits (including mean corpuscular volume, mean corpuscular hemoglobin concentration, and RBC count) (*Sankaran et al., 2013*; *van der Harst et al., 2012*). ETO2 (CBFA2T3) is a part of the erythroid transcription factor complex containing TAL1 and is required for expansion of erythroid progenitors (*Goardon et al., 2006*). Both stem cell factor receptor KIT and erythropoietin receptor (EPOR) mediated signaling are essential for erythropoiesis. Our screen identified KIT as one of the factors underlying common genetic variation. CCND3 fills a critical role in regulating the number of cell divisions during terminal erythropoiesis and has been validated as a causal gene associated with variation in RBC counts and size (*Sankaran et al., 2012*; *Ulirsch et al., 2019*).

Interacting networks of hits also emerged in other aspects of red blood cell differentiation and function. One of these centered around membrane and structural cytoskeletal proteins. Our method recovered characteristic RBC genes like solute carrier family 4 member 1 (SLC4A1), also known as band 3, (*Peters et al., 1996*), which serves as a key component of the RBC membrane skeleton. Likewise, it recovered a direct interacting partner for SLC4A1, ankyrin 1 (ANK1), which anchors the cytoskeleton and cell membrane (*Bennett and Stenbuck, 1979*), as well as N-ethylmaleimide Sensitive Factor, vesicle fusing ATPase (NSF), which facilitates membrane vesicle trafficking within the cell (*Glick and Rothman, 1987*).

Within the realm of mRNA translation, a number of genes emerged as hits that specifically highlight the role of the ribosome. This is interesting in light of recent work that has begun to illuminate erythroid-specific effects of ribosomal perturbations (*Khajuria et al., 2018*; *Ludwig et al., 2014*), although a connection between translation and common genetic variation affecting RBC traits has

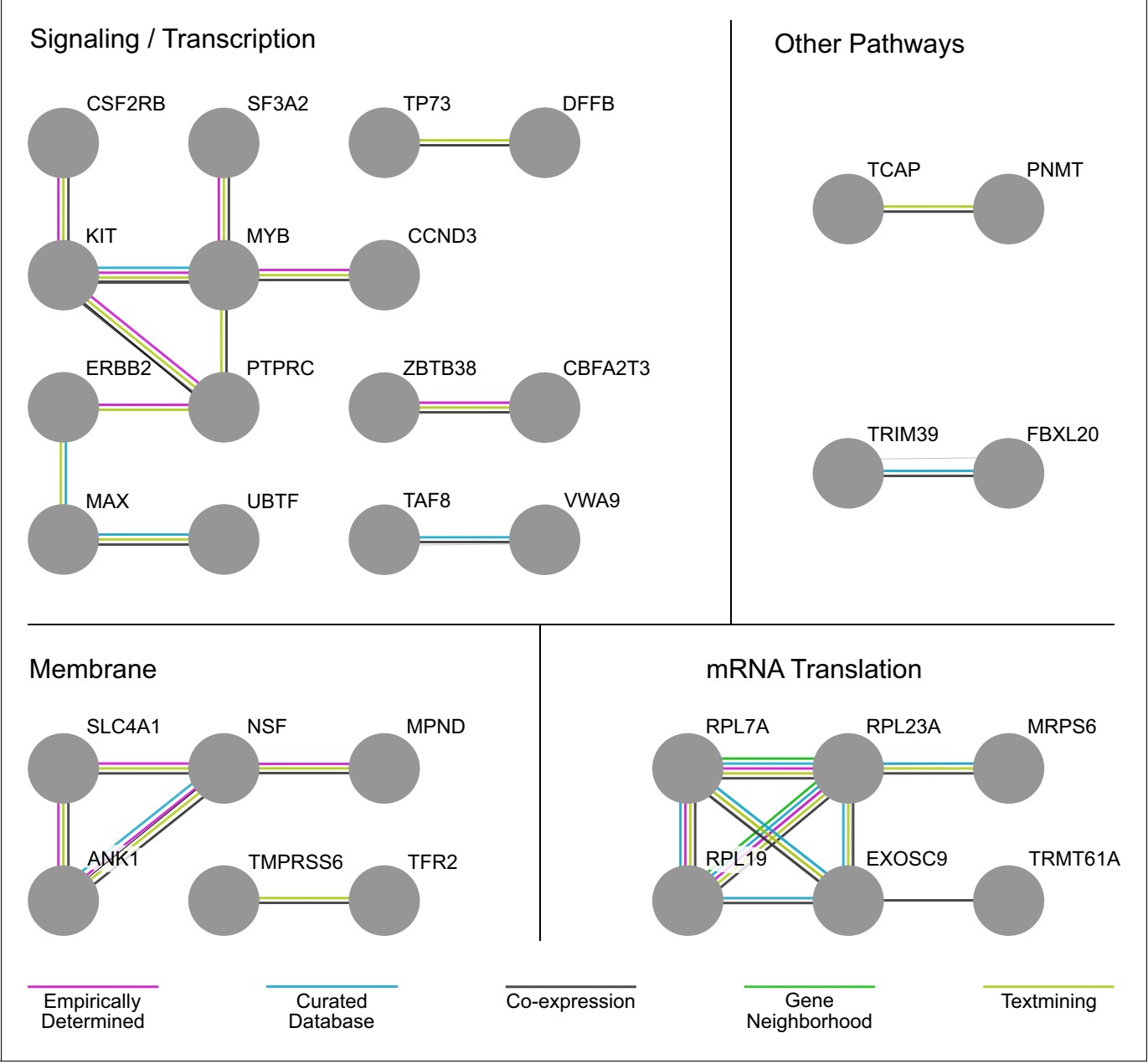

**Figure 4.** Analysis of Interactions Among Members of the Hit Set Identifies Signaling/Transcription, Membrane, and mRNA Translation-Related Subnetworks Important to Erythropoiesis. STRING interaction network analysis identifies signaling/transcription, membrane, and mRNA translation-related subnetworks important to erythropoiesis embedded in the genes identified in the screen hit set. Edges connecting the network are color-coded according to the evidence supporting the interaction. In STRING, this evidence can derive from empirical determination, curation in a database, co-expression of the respective gene nodes, genomic proximity, and text-mining of published literature.

DOI: https://doi.org/10.7554/eLife.44080.016

not been previously appreciated. Both *RPL7A* and *RPL19*, for instance, have been implicated by mutations observed in studies of Diamond-Blackfan anemia (*Gazda et al., 2012*; *DBA Group of Société d'Hématologie et d'Immunologie Pédiatrique-SHIP et al., 2012*). The common genetic variation affecting these ribosomal protein genes might contribute to the incomplete penetrance and variable expressivity of anemia seen in Diamond-Blackfan anemia patients (*Ulirsch et al., 2018*). Similar effects have been reported in neurodevelopmental disorders, where common genetic variants may influence phenotypic outcomes in patients (*Niemi et al., 2018*). Non-ribosomal hits in the

mRNA metabolism space were also found with both previously established and unknown ties to erythroid-specific phenotypes. Exosome component 9 (EXOSC9), for instance, has been suggested to act as part of the exosome complex as a specific gatekeeper of terminal erythroid maturation (*McIver et al., 2014*). Other unappreciated components, including the tRNA methyltransferase TRMT61A, also were highlighted through this analysis.

## Transferrin receptor 2 is a negative regulator of human erythropoiesis

We selected several candidate genes identified by our screen for further validation, given their previously unappreciated roles in human hematopoiesis/erythropoiesis. The first, transferrin receptor 2 (*TFR2*), encodes a protein canonically involved in iron homeostasis that has recently been shown to also regulate EPO receptor signaling (*Forejtnikovà et al., 2010*; *Nai et al., 2015*). Although TFR2 has been studied in the context of murine erythropoiesis, its role in human erythropoiesis has not been assessed. To validate TFR2 as a regulator of human erythropoiesis, we performed individual knockdown experiments using lentiviral shRNAs in primary human CD34$^+$ HSPCs undergoing erythroid differentiation. Significant knockdown of TFR2 was observed at both the mRNA (*Figure 5A*) and protein levels (*Figure 5B*) using two independent shRNAs from among the six targeting TFR2 in the screen. Though two of the six were outliers, the two chosen here for follow-up were part of the consensus group of four showing similar effects. Downregulation of TFR2 increased erythroid differentiation as observed by increased expression of erythroid-specific cell surface markers CD235a and CD71 at day 9 (shLUC ~22%; TFR2 sh1 ~42%; TFR2 sh2 ~40%) and day 12 of culture (shLUC ~60%; TF2 sh1 ~80%; TFR2 sh2 ~80%) (*Figure 5C and E*). Downregulation of TFR2 also improved the later stages of erythroid differentiation/maturation, as observed by loss of cell surface marker CD49d at day 15 of culture, an increased rate of enucleation, and through assessment of cell morphology (*Figure 5D* and *Figure 5—figure supplement 1A and B*). Previous studies have reported the isolation of TFR2 as a component of the erythropoietin (EPO) receptor complex (*Forejtnikovà et al., 2010*). To test if downregulation of TFR2 can result in increased EPO signaling *Kim et al. (2017)*, we measured EPO-dependent STAT5 phosphorylation after TFR2 knockdown in UT7/EPO cells (*Figure 5—figure supplement 1C*). TFR2 downregulation resulted in significantly higher pSTAT5 phosphorylation in comparison to the control with EPO stimulation from 0.02 U/mL to 200 U/mL (*Figure 5F*). In addition, the maximal pSTAT5 response could be achieved within a shorter period of EPO stimulation upon TFR2 downregulation (*Figure 5—figure supplement 1D*). Given our findings that TFR2 is a negative regulator of EPO signaling, it may be an ideal therapeutic target for conditions characterized by ineffective erythropoiesis like β-thalassemia (*Rund and Rachmilewitz, 2005*). A recent study has supported this hypothesis, showing that Tfr2 downregulation is beneficial in a mouse model of β-thalassemia (*Artuso et al., 2018*).

## SF3A2 is a key regulator of human erythropoiesis and is a disease modifier in a murine model of myelodysplastic syndrome

Extensive mRNA splicing occurs during the terminal stages of erythropoiesis (*Pimentel et al., 2016*). However, key regulators of this process remain largely undefined. Our study uncovered splicing factor 3A subunit 2 (SF3A2) in the subnetwork of erythropoiesis signaling and transcription hits (*Figure 4*). SF3A2 specifically was associated with maximal hairpin drop out at day 12 (FDR = 0.005) – a later time point in erythropoiesis. SF3A2 is a component of the U2SNRP complex whose binding to the branch point is critical for proper mRNA splicing (*Gozani et al., 1996*; *Gozani et al., 1998*). Knockdown of SF3A2 in primary human CD34$^+$ HSPCs results in decreased cell numbers during erythroid differentiation starting from day 7 (*Figure 6A–C*). To measure early effects of SF3A2 and to exclude potential toxicity of puromycin selection, we replaced the puromycin resistance gene with a GFP encoding cDNA in the lentiviral shRNA constructs. We achieved similar infection (30–40% on day 6) at the early time points between controls (shLuc) and shRNAs targeting *SF3A2* (*Figure 6—figure supplement 1A*). During erythroid differentiation, we observed a reduction in GFP-expressing cells comparable to the decreased cell numbers seen with the puromycin resistant constructs (*Figure 6—figure supplement 1A*). Decreased cell numbers were associated with decreased erythroid differentiation as measured by erythroid surface markers CD71 and CD235a (*Figure 6D*). We also observed an increase in non-erythroid lineages based on surface marker expression of CD11b (myeloid) and CD41a (megakaryocytic) (*Figure 6—figure supplement 1B*).

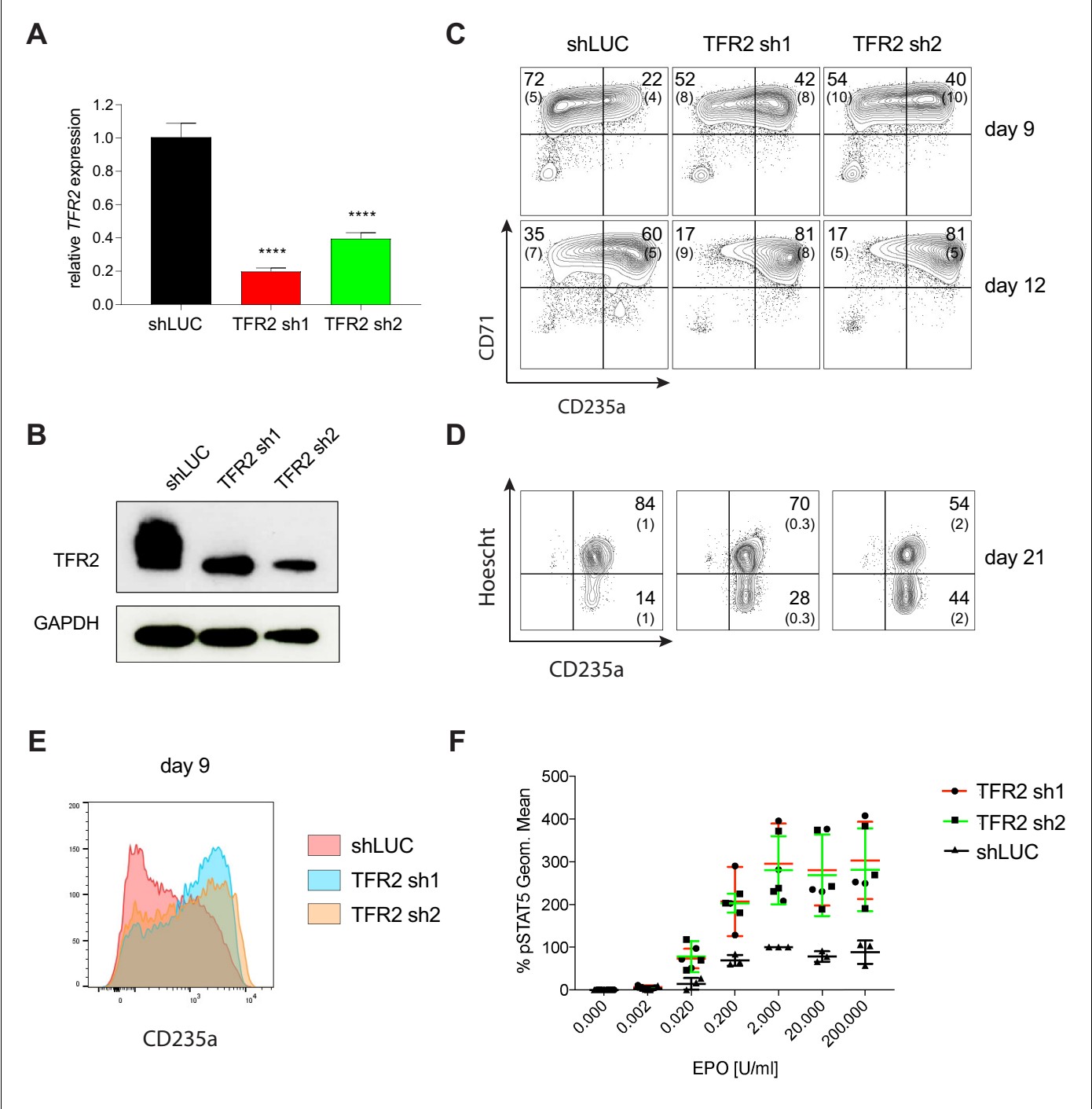

**Figure 5.** Transferrin receptor two is a Negative Regulator of Human Erythropoiesis. (**A**) Quantitative RT-PCR and (**B**) Western blot showing the expression of TFR2 in human CD34[+] cells five days post-infection with the respective lentiviral shRNAs targeting TFR2 (TFR2 sh1 and sh2) and a control luciferase gene (shLUC). (**C**) Representative FACS plots of erythroid cell surface markers CD71 (transferrin receptor) and CD235a (Glycophorin A) expression at various time points during erythroid differentiation. Percentages in each quadrant are represented as mean and standard deviation of 3 independent experiments (**D**) Hoechst staining showing more enucleated cells after TFR2 knockdown at day 21 of erythroid culture. (**E**) Representative histogram plots showing increased expression of CD235a (Glycophorin A) after TFR2 knockdown (**F**) Enhanced pSTAT5 response after TFR2 knockdown in UT7/EPO cells.

DOI: https://doi.org/10.7554/eLife.44080.017

The following figure supplement is available for figure 5:

**Figure supplement 1.** Additional Analysis Showing Transferrin Receptor two is a Negative Regulator of Human Erythropoiesis.

*Figure 5 continued on next page*

*Figure 5 continued*

DOI: https://doi.org/10.7554/eLife.44080.018

To identify the molecular mechanisms underlying the reduced differentiation of erythroid cells, we sorted stage-matched CD71$^+$/CD235a$^+$ cells and performed RNA-Seq analysis. We also ran this analysis in parallel for data from hematopoietic progenitors of patients with myelodysplastic syndrome (MDS), a disorder well-known for significant impairments in terminal erythropoiesis, either with or without somatic mutations in the related splicing factor *SF3B1* (*Obeng et al., 2016*). Cells treated with shRNA to suppress SF3A2 were found to differentially express 6061 genes with an adjusted p value < 0.05 as compared to the shLuc control, whereas only 807 genes were differentially expressed given the same threshold cutoff in the MDS patients with an *SF3B1* mutations compared to those without (*Figure 6E*). Genes from both the SF3A2 differentially expressed set and the SF3B1 differentially expressed set were significantly enriched for structural constituents of the ribosome (p<3.2×10$^{-44}$ and p<7.5×10$^{-24}$, respectively) among other cellular components and functions (*Figure 6—source data 3* and *4*). Examining differential splicing in the set of genes not differentially expressed in either condition, both were found to exhibit a similar proportion of altered splicing events, including alternative 3' splice sites, alternative 5' splice sites, mutually exclusive exons, and skipped exons (*Figure 6E*).

We therefore wanted to further explore this connection between SF3A2 and its role in common variation in RBC traits with SF3B1 and the role it plays in the pathogenesis of MDS. To this end, we utilized a recently developed faithful mouse model harboring the *Sf3b1*$^{K700E}$ mutation that displays characteristic features of MDS, including an anemia due to impaired erythropoiesis (*Obeng et al., 2016*). We tested if downregulation of Sf3a2 could worsen the already impaired erythropoiesis seen in these animals. Equal numbers of lineage-negative HSPCs were isolated from bone marrow of wild-type and *Sf3b1*$^{K700E}$ mice and infected with shRNAs targeting Sf3a2 and then erythroid differentiation was induced (*Figure 6—figure supplement 2A,B*). Consistent with previous reports, we observed that *Sf3b1*$^{K700E}$ cells show reduced erythroid differentiation and cell growth compared to wild-type cells infected with control non-targeting shRNAs (*Figure 6F,G*, *Figure 6—figure supplement 2C–E*). Downregulation of Sf3a2 using two independent shRNAs further worsens the defects in both erythroid differentiation and cell growth observed for *Sf3b1*$^{K700E}$ cells (*Figure 6F,G*, *Figure 6—figure supplement 2C–E*). This data suggest that modulation of SF3A2 could modify the alterations of erythropoiesis observed in the setting of somatic *SF3B1* MDS-causal mutations. This form of MDS is characterized by significant variation in the degree of anemia found at the time of presentation (*Chronic Myeloid Disorders Working Group of the International Cancer Genome Consortium et al., 2011*). We therefore attempted to examine whether such common genetic variation could contribute to such phenotypic variation. We identified a coding SNP, rs25672, in LD with the sentinel SNP at the locus, rs2159213 (r$^2$ = 0.737675 in CEU 1000 Genomes phase 3), in which *SF3A2* was the only gene identified by the linear mixed model as a hit. Prevalence of the alternate 'G' allele (which corresponds to the prevalence of the 'C' effect for *SF3A2*) is correlated with an increase in hemoglobin levels () that was likely insignificant due to the limited number of patients studied here. Unfortunately, larger cohorts in such a relatively rare disorder could not be identified. However, these findings suggest that the subtle variation noted in populations at the rs2159213 locus containing *SF3A2* may more profoundly cause variation among individuals with an acquired blood disorder, such as MDS, illustrating the value of such a gene-centric study to identify potential disease modifiers.

## Discussion

A major challenge in moving from GWAS-nominated variants to function is to identify potential target genes systematically. While many functional follow up approaches focus on causal variants, we reasoned that a gene-centered approach may be complementary to other emerging methods and represent a scalable approach for gaining broad insights into GWAS. To this end, we designed and executed a GWAS-informed high-throughput loss-of-function screen to identify key players in primary human HSPCs undergoing erythroid differentiation. Such dynamic in vitro systems afford a unique window through which to longitudinally screen, enabling unique insights to be gained into

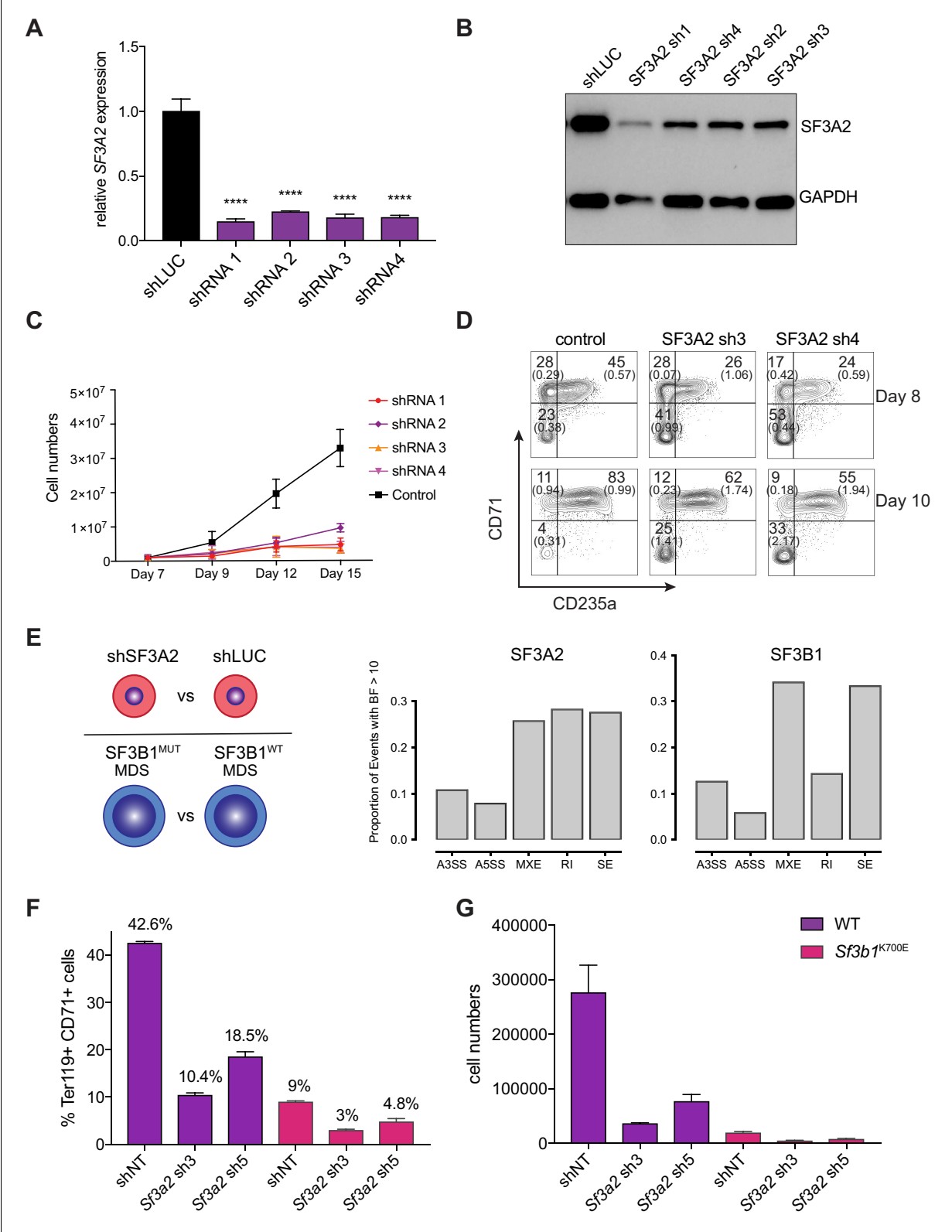

**Figure 6.** SF3A2 is a Key regulator of Human Erythropoiesis and Modulates Erythropoiesis Defects in a Murine Model of MDS. (**A**) Quantitative RT-PCR and (**B**) Western blot showing the expression of SF3A2 in human CD34+ cells five days post-infection with the respective lentiviral shRNAs targeting SF3A2 (sh1-4) and a control luciferase gene (shLUC). (**C**) Growth curves showing that downregulation of SF3A2 results in reduced total cell numbers during erythroid differentiation from three independent experiments. (**D**) Representative FACS plots of erythroid cell surface markers CD71 (transferrin

*Figure 6 continued on next page*

*Figure 6 continued*

receptor) and CD235a (Glycophorin A) expression at various time points during erythroid differentiation. Percentages in each quadrant are represented as mean and standard deviation of three independent experiments (E) Altered splicing events identified by RNA-Seq analysis of stage matched erythroid cells (shSF3A2 vs. shLUC). Overlapping changes observed in *SF3B1* mutant BM cells from MDS patients (Obeng et al) (*Figure 6—source data 5* and *6*). Differentially expressed genes and pathway analysis available in *Figure 6—source data 1–4*. (F) Lineage negative bone marrow cells from wildtype (WT) and *Sf3b1*[K700E] mice were infected with shRNAs targeting murine *Sf3a2* gene co-expressing a reporter GFP gene. Percentage of Ter119[+] CD71[+] erythroid cells within the GFP compartment after 48 hr in erythroid differentiation. (G) Total cell numbers of GFP[+] erythroid cells after 48 hr in erythroid differentiation.

DOI: https://doi.org/10.7554/eLife.44080.019

The following source data and figure supplements are available for figure 6:

**Source data 1.** Table containing the DESeq2 output for differentially expressed genes in cells undergoing SF3A2 knockdown or control shRNA treatment.
DOI: https://doi.org/10.7554/eLife.44080.022
**Source data 2.** Table containing the DESeq2 output for differentially expressed genes in MDS patients with and without mutations in *SF3B1*.
DOI: https://doi.org/10.7554/eLife.44080.023
**Source data 3.** Tables containing the GO component (Table 1) and function (Table 2) enrichments calculated using GOrilla for cells undergoing SF3A2 knockdown or control shRNA treatment.
DOI: https://doi.org/10.7554/eLife.44080.024
**Source data 4.** Tables containing the GO component (Table 1) and function (Table 2) enrichments calculated using GOrilla for MDS patient samples with and without mutations in *SF3B1*.
DOI: https://doi.org/10.7554/eLife.44080.025
**Source data 5.** Tables containing the differential splicing analysis for cells undergoing SF3A2 knockdown or control shRNA treatment.
DOI: https://doi.org/10.7554/eLife.44080.026
**Source data 6.** Tables containing the differential splicing analysis for MDS patient patient samples with and without mutations in *SF3B1*.
DOI: https://doi.org/10.7554/eLife.44080.027
**Figure supplement 1.** Additional Analysis Showing SF3A2 is Required for Human Erythropoiesis.
DOI: https://doi.org/10.7554/eLife.44080.020
**Figure supplement 2.** Additional Analysis of Erythropoiesis Defects Observed in *Sf3b1*[K700E] Murine Erythroid Cells upon SF3A2 knockdown.
DOI: https://doi.org/10.7554/eLife.44080.021

inherently non-stationary biological processes like erythropoiesis. The screen identified 77 gene hits at 38 of the original 75 loci used to design the library. Collectively, these hits had strongly amplified essentiality in erythroid cell lines, included a significant proportion of known, genetically-linked 'gold standard' erythroid genes, and were enriched for red blood cell trait-associated coding variants orthogonally identified through genetic fine-mapping. From a holistic perspective, the network of interacting gene hits highlighted a number of high-level biological components and pathways important for erythropoiesis, including specific signaling and transcription factors, membrane and structural components, and components involved in mRNA translation. It is interesting to note that the hits identified by our screen did not overlap those that would be identified with eQTLs from whole blood, which emphasizes how studies of variation in developing hematopoietic cells may not be accurately reflected by studies of circulating blood cells. It does, however, lend credence to the notion that there are complementary insights to be gained through these differing methods.

Functional follow-up on *SF3A2* and *TFR2*, two gene hits identified in the screen, were fruitful in elucidating mechanistic ties between alteration in mRNA splicing and EPO signaling activity, respectively, to observed perturbation of erythroid phenotypes. In addition, our studies suggest that at least SF3A2, and potentially other regulators such as some implicated mRNA translation factors, may be key disease modifiers that alter the impaired erythropoiesis seen in diseases like MDS or Diamond-Blackfan anemia. These outcomes strongly recommend further exploration of this approach as a rapid means to screen for genes underlying human erythroid differentiation, with the potential to connect back and explain the phenotypic links in GWAS studies. Moreover, since shRNA-based loss-of-function screens are readily accessible and offer demonstrated compatibility with primary cell model systems, we believe this approach provides a method that is portable and can be applied across a variety of lines of biological inquiry.

However, it is not a universal solution, and there are certainly a number of considerations that must be kept in mind regarding the extent to which this type of assay can be adopted across other diseases and traits. We acknowledge for it to be useful to a given research question, a suitable system capable of modeling the trait/disease of interest must first exist, and for many cellular systems

this is often challenging. Fortunately, this is a shortcoming that will diminish over time as our understanding of human biology and our ability to faithfully recapitulate in vivo microenvironments and processes improves, though this may be a distant prospect for exquisitely complex tissues like the brain or for traits/diseases that involve a larger number of cell types/interactions. Likewise, the use of shRNAs as the vehicle for perturbation carries with it unique challenges, chief among them the proclivity of shRNA to exert confounding off-target effects when compared to CRISPR-based methods. While this is true and unavoidable, the inclusion of appropriate controls, both at the experimental level and in modeling off-target contributors to observed phenotypic effects, provide an effective means to address this issue (*Tsherniak et al., 2017*). We chose to perform our screen in primary hematopoietic cells and thus were partially limited experimentally to the use of shRNA-based suppressive approaches. Finally, evidence has recently been published that the targets of identified non-coding variants are occasionally not within linkage disequilibrium blocks in the genome (*Whalen and Pollard, 2019*). This does not necessarily conflict with our results, since we identify hits at only 38 of 75 examined loci and provides an intriguing direction for further work that may elucidate how genetic and epigenomic structural blocks in the human genome can provide complementary information.

Our data show that gene-centric screens are valuable for GWAS follow-up. They are not limited to red cell traits and may be useful for other human traits/diseases, as has begun to be shown in diseases like type 2 diabetes (*Thomsen et al., 2016*). Data from such screens can be integrated with complementary insights gleaned from variant centric screens. Ultimately this could accelerate our understanding of human hematopoiesis and other biological processes, and aid in the development of applicable therapies.

# Materials and methods

**Key resources table**

| Reagent type (species) or resource | Designation | Source or reference | Identifiers | Additional information |
|---|---|---|---|---|
| Biological sample (Homo sapiens) | CD34 + mobilized peripheral blood | Fred Hutchinson Cancer Research Center | | |
| Cell line (Homo sapiens) | UT-7/EPO | NA | RRID:CVCL_5202 | maintained in Sankaran laboratory |
| Cell line (Mus musculus) | MEL | NA | | maintained in Sankaran laboratory |
| Genetic reagent (Mus musculus) | *Sf3b1*^K700E | *Obeng et al., 2016* | | Dr. Benjamin L. Ebert (Brigham Women's Hospital, Boston MA) |
| Recombinant DNA reagent (lentiviral shRNA) | PLKO.1-Puro (plasmid) | Sigma-Aldrich | RRID:Addgene_10878 | Pol III based shRNA backbone |
| Recombinant DNA reagent (lentiviral shRNA) | PLKO-GFP (plasmid) | this paper | | GFP version of pLKO.1-Puro |
| Recombinant DNA reagent (lentiviral shRNA) | SFFV-Venus-mir30 shRNA (plasmid) | this paper | | Pol II based shRNA backbone |
| Antibody | mouse monoclonal anti-human CD235a-APC | Thermo Fisher Scientific | Cat#: 17-9987-42; RRID:AB_2043823 | FACS (5 ul per test) |
| Antibody | mouse monoclonal anti-human CD71-FITC | Thermo Fisher Scientific | Cat#: 11-0719-42; RRID:AB_1724093 | FACS (5 ul per test) |
| Antibody | mouse monoclonal anti-human CD71-PEcy7 | Thermo Fisher Scientific | Cat#: 25-0719-42; RRID:AB_2573366 | FACS (5 ul per test) |

*Continued on next page*

*Continued*

| Reagent type (species) or resource | Designation | Source or reference | Identifiers | Additional information |
|---|---|---|---|---|
| Antibody | mouse monoclonal ant-human CD49d-PE | Miltenyi Biotec | Cat#: 130-093-282; RRID:AB_1036224 | FACS (10 ul per test) |
| Antibody | mouse monoclonal anti-human CD41a-PE | Thermo Fisher Scientific | Cat#: 12-0419-42; RRID:AB_10870785 | FACS (5 ul per test) |
| Antibody | mouse monoclonal anti-human CD11b-PE | Thermo Fisher Scientific | Cat#: 12-0118-42; RRID:AB_2043799 | FACS (5 ul per test) |
| Antibody | Rat monoclonal anti-mouse Ter119-APC | Thermo Fisher Scientific | Cat#: 17-5921-82; RRID:AB_469473 | FACS (0.25 ug/test) |
| Antibody | Rat monoclonal anti-mouse CD71-PE | Thermo Fisher Scientific | Cat#: 12-0711-82; RRID:AB_465740 | FACS (0.5 ug/test) |
| Antibody | mouse monoclonal anti-phospho STAT5 Alexa Fluor-647 | BD Bioscience | Cat: 612599; RRID:AB_399882 | FACS (1:20) |
| Antibody | mouse monoclonal anti-GAPDH | Santa Cruz Biotechnology | sc-32233; RRID:AB_627679 | Western (1:20,000) |
| Antibody | mouse monoclonal anti-TFR2 | Santa Cruz Biotechnology | sc-32271; RRID:AB_628395 | Western (1:200) |
| Antibody | mouse monoclonal anti-SF3A2 | Santa Cruz Biotechnology | sc-390444 | Western (1:1000) |
| Sequence-based reagent | shLUC | Sigma-Aldrich | TRCN0000072259 | 5'- CGCTGAGTACT TCGAAATGTC-3' |
| Sequence-based reagent | TFR2 sh1 (human) | Sigma-Aldrich | TRCN0000063628 | 5'-GCCAGATCACT ACGTTGTCAT-3' |
| Sequence-based reagent | TFR2 sh2 (human) | Sigma-Aldrich | TRCN0000063632 | 5-CAACAACATCT TCGGCTGCAT-3' |
| Sequence-based reagent | SF3A2 sh1 (human) | Sigma-Aldrich | TRCN0000000060 | 5'-CTACGAGACCAT TGCCTTCAA-3' |
| Sequence-based reagent | SF3A2 sh2 (human) | Sigma-Aldrich | TRCN0000000061 | 5'-CCTGGGCTCCT ATGAATGCAA-3' |
| Sequence-based reagent | SF3A2 sh3 (human) | Sigma-Aldrich | TRCN0000000062 | 5'-CAAAGTGACC AAGCAGAGAGA-3' |
| Sequence-based reagent | SF3A2 sh4 (human) | Sigma-Aldrich | TRCN0000000063 | 5'-ACATCAACAAG GACCCGTACT-3' |
| Commercial assay or kit | RNeasy Mini Kit | QIAGEN | Cat#: 74104 | |
| Commercial assay or kit | iScript cDNA synthesis Kit | Bio-Rad | Cat#: 1708891 | |
| Commercial assay or kit | iQ SYBR Green Supermix | Bio-Rad | Cat#: 170–8882 | |
| Commercial assay or kit | NucleoSpin Blood XL-Maxi kit | Clonetch | Cat#: 740950.1 | |
| Commercial assay or kit | Lineage Cell Depletion Kit (mouse) | Miltenyi | Cat#: 130-090-858 | |
| Commercial assay or kit | Nextera XT DNA Library Preparation Kit | Illumina | Cat#: FC-131–1096 | |

*Continued on next page*

*Continued*

| Reagent type (species) or resource | Designation | Source or reference | Identifiers | Additional information |
|---|---|---|---|---|
| Commercial assay or kit | NextSeq 500/550 High Output Kit v2.5 (75 Cycles) | Illumina | Cat#: 20024906 | |
| Commercial assay or kit | Bioanalyzer High Sensitivity DNA Analysis | Agilent | Cat#: 5067–4626 | |
| Commercial assay or kit | Agencourt AMPure XP | Beckman-Coulter | Cat#: A63881 | |
| Commercial assay or kit | TaKaRa Ex Taq DNA Polymerase | Takara | Cat#: RR001B | |
| Commercial assay or kit | Qubit dsDNA HS Assay Kit | Thermo Fisher | Cat#: Q32854 | |
| Chemical compound, drug | Human Holo-Transferrin | Sigma Aldrich | Cat#: T0665-1G | |
| Peptide, recombinant protein | Humulin R (insulin) | Lilly | NDC 0002-8215-01 | |
| Peptide, recombinant protein | Heparin | Hospira | NDC 00409-2720-01 | |
| Peptide, recombinant protein | Epogen (recombinant erythropoietin) | Amgen | NDC 55513-267-10 | |
| Peptide, recombinant protein | Recombinant human stem cell factor (SCF) | Peprotech | Cat#: 300–07 | |
| Peptide, recombinant protein | Recombinant human interleukin-3 (IL-3) | Peprotech | Cat#: 200–03 | |
| Peptide, recombinant protein | Recombinant mouse stem cell factor (SCF) | R&D systems | Cat# 455-MC-010 | |
| Peptide, recombinant protein | recombinant mouse Insulin like Growth Factor 1 (IGF1) | R&D systems | Cat# 791 MG-050 | |
| Chemical compound, drug | Hoechst 33342 | Life Technologies | Cat#: H1399 | FACS (1:1000) |
| Chemical compound, drug | Fixation Buffer | BD Bioscience | Cat#: 554655 | |
| Chemical compound, drug | Perm Buffer III | BD Bioscience | Cat#: 558050 | |
| Chemical compound, drug | May-Grünwald Stain | Sigma-Aldrich | Cat#: MG500 | |
| Chemical compound, drug | Giemsa Stain | Sigma-Aldrich | Cat#: GS500 | |
| Software, algorithm | STAR | *Dobin et al., 2013* | RRID:SCR_015899 | |
| Software, algorithm | MISO | *Katz et al., 2010* | RRID:SCR_003124 | |
| Software, algorithm | R | The R Foundation | RRID:SCR_001905 | |
| Software, algorithm | Salmon | *Patro et al., 2017* | RRID:SCR_017036 | |
| Software, algorithm | GOrilla | *Eden et al., 2009* | RRID:SCR_006848 | |

*Continued on next page*

*Continued*

| Reagent type (species) or resource | Designation | Source or reference | Identifiers | Additional information |
|---|---|---|---|---|
| Software, algorithm | VEP | *McLaren et al., 2016* | RRID:SCR_007931 | |
| Software, algorithm | FlowJo version 10 | FlowJo | RRID:SCR_008520 | |
| Software, algorithm | GraphPad Prism 7 | GraphPad Software Inc | RRID:SCR_002798 | |
| Software, algorithm | Python 2, 3 | Python Software Foundation | RRID:SCR_008394 | |
| Software, algorithm | PLINK | *Chang et al., 2015* | RRID:SCR_001757 | |
| Software, algorithm | PoolQ | Broad Institute | https://portals.broadinstitute.org/gpp/public/software/poolq | |

## Design of the shRNA library

PLINK version 1.9 and 1000 genomes phase one data were utilized to expand 75 SNPs previously identified in a RBC trait GWAS to include a genomic region in linkage disequilibrium with $r^2 \geq 0.8$. Each of these regions was then further expanded to the nearest recombination hotspot. All genes in the Ensembl assembly GRCh37 were expanded to include 110 kb upstream and 40 kb downstream of the transcription start and end sites, respectively, to maximize capture of non-coding regulatory interactions, based upon previously published observations. Genes with windows calculated in this way found to be overlapping with any of the SNP windows were flagged for inclusion in the screen. In addition, each locus was examined individually, and in cases of gene deserts, unusually proximal recombination hotspots, or other unusual genomic structures, the SNP region was expanded to include additional genes nearby. This resulted in a total of 389 test genes, which were each targeted by 4–7 distinct shRNAs. Also included in the library were shRNAs targeting a set of 8 validated erythroid genes (*GATA1, RPL5, RPS19, EPOR, ALAS2, CDAN1, SEC23B, ZFPM1*). A pooled library of 2803 TRC clones was produced from the sequence-validated TRC shRNA library (*Moffat et al., 2006*) and included shRNAs targeting control genes and essential genes.

## Pooled shRNA screening

Mobilized peripheral blood CD34$^+$ cells from three separate donors (7 ~ 11 * 10$^6$ cells per donor) were differentiated into erythroid cells using a three-stage system that has been previously described (*Hu et al., 2013*). Cells were cultured using IMDM containing 2% human plasma, 3% human AB serum, 200 µg/ml human holo-transferrin, 3 IU/mL heparin, and 10 mg/mL insulin (base medium). During days 0 to 7, cells were supplemented with IL-3 (1 ng/mL), SCF (10 ng/ml), and EPO (3 IU/ml). On day 2 of this culture, cells were transduced with the pooled lentiviral shRNA library prepared by Broad Institute Genetic Perturbation Platform (1 ml of virus per 0.75 * 10$^6$ cells) by spinfection at 2000 rpm for 90 min with 6 µg/ml polybrene. During days 7 to 13, cells were supplemented with SCF and EPO only. After day 13, cells were supplemented with EPO alone and the holo-transferrin concentration was increased to 1 mg/ml. A minimum of 10 * 10$^6$ cells was re-plated at each time point to ensure appropriate library representation and prevent bottlenecks among the infected cells. Cell pellets were made from 20 ~ 80 * 10$^6$ cells at days 4, 6, 9, 12, 14, and 16. At the conclusion of the pooled screen, genomic DNA (gDNA) was extracted from the cell pellets using Nucleo-Spin Blood XL-Maxi kit (Clonetech) according to kit specifications. The shRNA-containing region was PCR amplified from the purified gDNA and barcoded using the following conditions: 0.5 µl P5 primer mix (100 µM), 10 µl P7 primer mix (5 µM), 8 µl dNTP mix, 1x ExTaq buffer, 1.5 µl of ExTaq DNA polymerase (Takara), and up to 10 µg genomic DNA in a total reaction volume of 100 µl. A total of 40 ~ 87.5 µg gDNA was used as template from each condition. Thermal cycler PCR conditions consisted of heating samples to 95℃ for 5 min; 28 cycles of 95℃ for 30 s, 53℃ for 30 s, and 72℃ for 20 s; and 72℃ for 10 min. Equal amounts of samples were then mixed and purified using

AMPure XP for PCR purification (Beckman Coulter). Samples were sequenced using a custom sequencing primer using standard Illumina conditions by the Broad Institute Genetic Perturbation Platform. Sequencing reads were deconvolved and hairpin counts were quantified for subsequent analysis by counting against the barcode reference using PoolQ (https://portals.broadinstitute.org/gpp/public/dir/download?dirpath=software&filename=poolq-2.2.0-manual.pdf).

P5 primer

**AATGATACGGCGACCACCGAGATCT**ACACTCTTTCCCTACACGACGCTCTTCCGATCT[s]**TCTTGTGGAAAGG\*A\*C\*G\*A**

A mix of P5 primers with stagger regions [s] of different length was used to maintain sequence diversity across the flow-cell.

P7 primer

**CAAGCAGAAGACGGCATACGAGAT**NNNNNNNNNGTGACTGGAGTTCAGACGTGTGCTCTTCCGATCT**TCTACTATTCTTTCCCCTGCA\*C\*T\*G\*T**

Independently barcoded P7 primers was used for each condition.

NNNNNNNN – barcode region

## Analysis of the shRNA screen

A jupyter notebook, along with companion data files to reproduce the figures and analysis in this work, can be found at https://github.com/sankaranlab/shRNA_screen (*Nandakumar and McFarland, 2019*; copy archived at https://github.com/elifesciences-publications/shRNA_screen). To summarize, three separate donor primary CD34$^+$ cells populations were run as replicates in the shRNA screen. A pseudocount of 1 was added to all shRNA-encoding DNA count totals and these counts were subsequently normalized to counts per million (CPM) and $\log_2$ transformed. A linear mixed model was constructed to fit fixed effects for each gene at each time point since transfection (t) using the $\log_2$ fold change from initial hairpin counts as the response variable ($y$). A random effect was included to capture variations in efficacy and off-target effects for each shRNA ($h$) used to target a given gene that could accumulate over the course of the experiment. The resulting model, $y \sim t + (0 + t|h)$, was fit in R-3.4 using the lme4 package. Genes hits were called from the set of genes with β coefficient effect size >0.1 and the Wald chi-square test adjusted q value < 0.1. Enrichment of erythroid essential genes within the hit set was calculated by running 1 million permutations against the distribution of K562 essentiality for all genes included in the library, panels of genes nominated by sets of significant GWAS-associated lipid trait SNPs (*Global Lipids Genetics Consortium et al., 2013*), and against all genes in the genome (Ensembl GRCh37p9). Coding variants and protein effects (i.e. missense or nonsense) were annotated based on the Variant Effect Predictor software (https://www.ensembl.org/vep). Enrichment for identification of the included 5 'gold standard' genes and for red blood cell trait-associated coding variants were each accomplished using identical permutation schemes. Expression of the hit genes in various cell states/stages of differentiation was derived from the cited datasets and permuted across all unique stages to determine stage-specific enrichment. The interaction network surrounding the 77 hits identified in the screen was generated in the latest version of STRING (10.5) and filtered for the purposes of display to only those nodes with at least one edge to another node among the hits. For the comparison with eQTL-based methods of hit identification, we used the whole blood summary statistics from GTEx and intersected them with genomic regions in linkage disequilibrium $r^2$ >0.8 with the 75 sentinel SNPs from the van der Harst et al. study used to identify the library of genes targeted in the screen (see *Figure 1A*). These regions were padded to a fixed 100 kb, as many of the regions were small. This yielded 139 genes that one could argue would be nominated on an eQTL-basis from the total pool of 8661 genes with a whole blood eQTL. We performed a Fisher's exact test on the contingency table comparing hits from our method of nomination with the set of eQTLs and eQTL-nominated genes. There were 35 hits (of the 77 total in our screen) present among the whole eQTL set, but 0 hits found among the 139 genes nominated by using eQTLs from whole blood. The data used for the analyses described in this manuscript were obtained from the GTEx Portal on 06/01/18.

## RNA-Seq

Stage matched CD71$^+$/ CD235a$^+$ cells derived from CD34$^+$ HSPCs infected with SF3A2 sh3, sh4 and shLUC were FACS sorted at day 8 of erythroid differentiation. RNA was isolated using a RNAqueous Micro kit (Invitrogen) according to the manufacturer's instructions. DNase digestion was performed before RNA was quantified using a Qubit RNA HS Assay kit (Invitrogen). 1–10 ng of RNA were used as input to a modified SMART-seq2 (*Picelli et al., 2014*) protocol and after reverse transcription, 8–9 cycles of PCR were used to amplify transcriptome library. Quality of whole transcriptome libraries was validated using a High Sensitivity DNA Chip run on a Bioanalyzer 2100 system (Agilent), followed by library preparation using the Nextera XT kit (Illumina) and custom index primers according to the manufacturer's instructions. Final libraries were quantified using a Qubit dsDNA HS Assay kit (Invitrogen) and a high sensitivity DNA chip run on a Bioanalyzer 2100 system (Agilent). All libraries were sequenced using Nextseq High Output Cartridge kits and a Nextseq 500 sequencer (Illumina). Libraries were sequenced using 2 $\times$ 38 bp paired end reads.

## RNA-seq differential expression analysis

For differential expression analysis, paired end sequencing reads from our SF3A2 shRNA knockdown experiments and obtained from the SF3B1 mutant datasets (*Obeng et al., 2016*) were quantified using Salmon version 0.11.1 (*Patro et al., 2017*) with default parameters and an index constructed from Gencode annotations version 28. Differential expression of quantified counts was calculated using DESeq2 (*Love et al., 2014*) in R-3.4. Enrichment for functions and components of the cell among the differentially expressed gene sets were quantified using GOrilla (*Eden et al., 2007*; *Eden et al., 2009*).

## RNA-seq differential splicing analysis

Paired end sequencing reads from our SF3A2 shRNA knockdown experiments and obtained from the cited SF3B1 mutant datasets were aligned using STAR version 2.5.2 in two-pass mode. Differential splicing was quantified using MISO version 0.5.4 in Python 2.7 using the instructions and annotation files provided with the package (*Katz et al., 2010*). The software's default cutoff of Bayes factor of 10 or greater was used to call differential splice forms.

## Analysis of hemoglobin levels for MDS patients with or without SF3A2 mutations

Genotyped MDS patient hemoglobin level measurements were obtained from the laboratory of J. Maciejewski. 1000GENOMES phase three data were used to find a SNP encoded in whole-exome sequencing data (rs25672) in high LD ($r^2$ = 0.737675) with the SF3A2-associated sentinel SNP (rs2159213). An ordinary least squares linear regression was used to fit the patient hemoglobin levels to the number of SF3A2 minor alleles present in each patient (log likelihood ratio test p=0.140).

## Phosphorylated STAT5 assessment with intracellular flow cytometry

UT-7/EPO cells were cultured in DMEM medium supplemented with 10% Fetal Bovine Serum and 2 U/mL EPO. 5 days post-infection with TFR2 shRNAs, UT-7/EPO cells were cytokine starved overnight. On the next day, cells were treated with EPO in a dose dependent manner ((0 U/mL, 0.002 U/mL, 0.02 U/mL, 0.2 U/mL, 2 U/mL, 20 U/mL and 200 U/mL) and incubated 37°C for 30 min. Alternatively the cells were treated with 2 U/ml EPO in a time dependent manner (15, 30, 60, 120,180 min). Treated cells were gently mixed with pre-warmed Fixation Buffer (BD Bioscience) at 37°C for 10 min to fix cells. To permeabilize cells for intracellular staining, cells were resuspended in pre-chilled Perm Buffer III (BD Bioscience) for 30 min at 4°C. After three washes with 3% FBS in PBS, samples were stained either with Alexa Fluor-647 Mouse Anti-phospho-STAT5 (pY694; 1:20 dilution) for 1 hr in the dark at room temperature. A BD Accuri C6 Cytometer (BD Bioscience) was used to acquire mean fluorescent intensity (MFI) of phospho-STAT5-Alexa Fluor 647. The MFI of phospho-STAT5-Alexa Fluor 647 of gated single cells was calculated using FlowJo (version 10.0.8r1). Unstimulated UT7/EPO cells were used as a negative control.

## May-Grünwald-Giemsa staining

Approximately 50,000–200,000 cells were harvested, washed once at 300 x g for 5 min, resuspended in 200 µL FACS buffer and spun onto poly-L-lysine coated glass slides (Sigma Aldrich) with a Shandon 4 (Thermo Fisher) cytocentrifuge at 300 rpm for 4 min. Visibly dry slides were stained with May-Grünwald solution for 5 min, rinsed four times for each 30 s in H2O, transferred to Giemsa solution for 15 min and washed as described above. Slides were dried overnight and mounted with coverslip. All images were taken with AxioVision software (Zeiss) at 100 x magnification.

## Mouse erythroid differentiation culture

Bone marrow cells that were isolated from $Sf3b1^{K700E\ +/-}$ mice and littermate controls were lineage depleted using the Lineage Cell Depletion Kit, mouse (Miltenyi Biotech) according to manufacturer's protocols. Lineage negative cells were immediately transduced with lentiviral shRNAs targeting SF3A2 or controls (MOI −90) by spinfection at 2000 rpm for 90 min. The cells were cultured in erythroid maintenance medium (StemSpan-SFEM; StemCell Technologies) supplemented with 100 ng/mL recombinant mouse stem cell factor (SCF) (R&D Systems), 40 ng/mL recombinant mouse IGF1 (R&D Systems), 100 nM dexamethasone (Sigma), and 2 U/mL erythropoietin (Amgen) and cultured at 37°C for 36 hr. Following this, the cells were cultured for another 48 hr in erythroid differentiation medium (Iscove modified Dulbecco's medium containing 15% (vol/vol) FBS (Stemcell), 1% detoxified BSA (Stemcell), 500 µg/mL holo-transferrin (Sigma-Aldrich), 0.5 U/mL Epoetin (Epo; Amgen), 10 µg/mL recombinant human insulin (Sigma-Aldrich), and 2 mM L-glutamine (Invitrogen)) at 37°C.

## Flow cytometry analyses and antibodies

All flow cytometry data were acquired using either using LSR II SORP or LSR Fortessa flow cytometers (BD Biosciences). All staining was carried out in FACS buffer (2% FBS in PBS) for 30 min on ice unless otherwise described. The following antibodies were used anti-human CD235a-APC (eBioscience, Clone HIR2), anti-human CD71-FITC (eBioscience, Clone OKT9), anti-human CD71-PEcy7 (eBioscience, Clone OKT9), ant-human CD49d-PE (Miltenyi, Clone MZ18-24A9), anti-human CD41a-PE (eBioscience, Clone HIP8), anti-human CD11b-PE (eBioscience, Clone ICRF44), anti-mouse Ter119-APC (eBioscience, Clone TER119), anti-mouse CD71-PE (eBioscience, Clone R17217) and Alexa Fluor-647 anti-phospho STAT5 (pY694) (BD Bioscience Cat#: 612599). Hoechst 33342 (Life Technologies, H1399) was used to visualize nuclei.

## shRNA sequences

The following lentiviral shRNA constructs were generated in Polymerase III based shRNA backbone pLKO.1-puro (Sigma Aldrich).

shLUC
5'-CCGGCGCTGAGTACTTCGAAATGTCCTCGAGGACATTTCGAAGTACTCAGCGTTTTTG-3'
TFR2 sh1
5'-CCGGGCCAGATCACTACGTTGTCATCTCGAGATGACAACGTAGTGATCTGGCTTTTTG-3
TFR2 sh2
5'-CCGGCAACAACATCTTCGGCTGCATCTCGAGATGCAGCCGAAGATGTTGTTGTTTTTG-3'
SF3A2 sh1 (human)
5'-CCGGCTACGAGACCATTGCCTTCAACTCGAGTTGAAGGCAATGGTCTCGTAGTTTTT-3
SF3A2 sh2 (human)
5'-CCGGCCTGGGCTCCTATGAATGCAACTCGAGTTGCATTCATAGGAGCCCAGGTTTTT-3'
SF3A2 sh3 (human)
5'-CCGGCAAAGTGACCAAGCAGAGAGACTCGAGTCTCTCTGCTTGGTCACTTTGTTTTT-3
SF3A2 sh4 (human)
5'-CCGGACATCAACAAGGACCCGTACTCTCGAGAGTACGGGTCCTTGTTGATGTTTTTT-3'

The following lentiviral shRNA constructs were generated in Polymerase II based mir30 shRNA backbone developed in the lab SFFV-Venus-mir30 shRNA backbone.

shNT(non-targeting)
5'_TGCTGTTGACAGTGAGCGATCTCGCTTGGGCGAGAGTAAGTAGTGAAGCCACAGATGTAC
TTACTCTCGCCCAAGCGAGAGTGCCTACTGCCTCGGA_3'
$Sf3a2$ sh1 (mouse)

5'_TGCTGTTGACAGTGAGCGCGGAGGTGAAGAAGTTTGTGAATAGTGAAGCCACAGATGTA TTCACAAACTTCTTCACCTCCATGCCTACTGCCTCGGA_3'

*Sf3a2* sh2 (mouse)
5'_TGCTGTTGACAGTGAGCGACCACCGTTTCATGTCTGCTTATAGTGAAGCCACAGATGTA TAAGCAGACATGAAACGGTGGCTGCCTACTGCCTCGGA_3'

*Sf3a2* sh3 (mouse)
5'_TGCTGTTGACAGTGAGCGATCCTGCCTTGAGCCTATTAAATAGTGAAGCCACAGATGTA TTTAATAGGCTCAAGGCAGGACTGCCTACTGCCTCGGA_3'

*Sf3a2* sh4 (mouse)
5'_TGCTGTTGACAGTGAGCGACCACTGGAACAGAGAAACCAATAGTGAAGCCACAGATGTA TTGGTTTCTCTGTTCCAGTGGGTGCCTACTGCCTCGGA_3'

*Sf3a2* sh5 (mouse)
5'_TGCTGTTGACAGTGAGCGATGGAGGTGAAGAAGTTTGTGATAGTGAAGCCACAGATGTA TCACAAACTTCTTCACCTCCACTGCCTACTGCCTCGGA_3'

## sgRNA sequences
The following sgRNA sequences targeting a variant in the Duffy promoter were cloned into SpCas9 and KRAB-dcas9 constructs.

Duffy sgRNA 3: 5'-GGCCCGCAGACAGAAGGGCT-3'
Duffy sgRNA 5: 5'-GGGCCATCAGGTTCTGGGCA-3'
Control sgRNA: 5'-ATCGCGAGGACCCGTTCCGCC-3'

## qPCR primers
*TFR2* Fwd: 5'-ATCCTTCCCTCTTCCCTCCC-3'
*TFR2* Rev: 5'-CCATCCAGCCACATGGTTCT-3
*SF3A2* Fwd: 5'-CCTGAGAAGGTCAAGGTGGA-3'
*SF3A2* Rev: 5'-CTCCGAGTCTCTCTGCTTGG-3'

## Western blot antibodies
Anti-GAPDH (Santa Cruz Biotechnology, sc-32233); anti-TFR2 (Santa Cruz Biotechnology, sc- sc-32271); anti-SF3A2 (Santa Cruz Biotechnology, sc-390444).

## Source data
Important data associated with figures in the manuscript are included below. For a full set of data-sets and resources used in the analyses, please see the companion GitHub repository (https://github.com/sankaranlab/shRNA_screen).

## Acknowledgements
We thank members of the Sankaran laboratory for valuable comments and suggestions on these studies. This work was supported by the National Institutes of Health grants R01 DK103794 and R33 HL120791, as well as the New York Stem Cell Foundation (to VGS). VGS is a New York Stem Cell Foundation-Robertson Investigator.

## Additional information

### Funding

| Funder | Grant reference number | Author |
| --- | --- | --- |
| National Institutes of Health | R33HL120791 | Vijay G Sankaran |
| New York Stem Cell Foundation | | Vijay G Sankaran |
| National Institutes of Health | R01DK103794 | Vijay G Sankaran |

The funders had no role in study design, data collection and interpretation, or the decision to submit the work for publication.

**Author contributions**
Satish K Nandakumar, Sean K McFarland, Conceptualization, Formal analysis, Validation, Investigation, Methodology, Writing—original draft, Writing—review and editing; Laura M Mateyka, Validation, Investigation, Writing—review and editing; Caleb A Lareau, Jacob C Ulirsch, Jesse M Engreitz, Bartlomiej Przychodzen, Jaroslaw P Maciejewski, Benjamin L Ebert, Formal analysis, Writing—review and editing; Leif S Ludwig, Gaurav Agarwal, Marie McConkey, Investigation, Writing—review and editing; Glenn S Cowley, John G Doench, Formal analysis, Investigation, Writing—review and editing; David E Root, Conceptualization, Formal analysis, Writing—review and editing; Vijay G Sankaran, Conceptualization, Resources, Supervision, Funding acquisition, Methodology, Writing—original draft, Project administration, Writing—review and editing

**Author ORCIDs**
Satish K Nandakumar (iD) https://orcid.org/0000-0002-7853-426X
Caleb A Lareau (iD) https://orcid.org/0000-0003-4179-4807
Jesse M Engreitz (iD) http://orcid.org/0000-0002-5754-1719
John G Doench (iD) https://orcid.org/0000-0002-3707-9889
Vijay G Sankaran (iD) https://orcid.org/0000-0003-0044-443X

**Decision letter and Author response**
Decision letter https://doi.org/10.7554/eLife.44080.040
Author response https://doi.org/10.7554/eLife.44080.041

## Additional files

**Supplementary files**
• Transparent reporting form
DOI: https://doi.org/10.7554/eLife.44080.028

**Data availability**
1000 Genomes human variation dataset: The 1000 Genomes Project Consortium. (2015) Recombinant hotspots access at: ftp://ftp-trace.ncbi.nih.gov/1000genomes/ftp/pilot_data/technical/reference/. Phase 1 data (for PLINK) accessed at: https://www.cog-genomics.org/plink/1.9/resources. Phase 3 data accessed at: http://www.internationalgenome.org/category/phase-3/. Pooled screen abundance data for shRNA targeting red blood cell trait GWAS-nominated genes during the course of in vitro differentiation of human CD34+ cells (SK Nandakumar, SK McFarland et al., 2019) is available on the project's companion GitHub repository: https://github.com/sankaranlab/shRNA_screen/tree/master/ref/shref.csv (copy archived at https://github.com/elifesciences-publications/shRNA_screen).

The following dataset was generated:

| Author(s) | Year | Dataset title | Dataset URL | Database and Identifier |
|---|---|---|---|---|
| Satish K Nandakumar, Sean K McFarland, Laura M Mateyka, Caleb A Lareau, Leif S Ludwig, Vijay G Sankaran | 2019 | Effects of shRNA knockdown of SF3A2 on splicing during human erythropoiesis | http://www.ncbi.nlm.nih.gov/geo/query/acc.cgi?acc=GSE129603 | NCBI Gene Expression Omnibus, GSE129603 |

The following previously published datasets were used:

| Author(s) | Year | Dataset title | Dataset URL | Database and Identifier |
|---|---|---|---|---|
| Yan H, Hale JP, Jaffray J, Li J, Wang Y, Huang Y, An X, Hillyer C, Wang N, Kinet S, Taylor N, Narla M, Narla A, Blanc L | 2018 | Human adult and fetal erythropoiesis gene expression | http://www.ncbi.nlm.nih.gov/geo/query/acc.cgi?acc=GSE107218 | NCBI Gene Expression Omnibus, GSE107218 |
| Corces MR, Buenrostro JD, Wu B, Greenside PG, Chan SM, Koenig JL, Snyder MP, Pritchard JK, Kundaje A, Greenleaf W, Majeti R, Chang H | 2016 | Human hematopoietic lineage gene expression | http://www.ncbi.nlm.nih.gov/geo/query/acc.cgi?acc=GSE74912 | NCBI Gene Expression Omnibus, GSE74912 |
| Global Lipids Genetics Consortium | 2013 | SNP sets identified by GWAS of LDL, HDL, and triglyceride traits | http://csg.sph.umich.edu/willer/public/lipids2013/ | Center for Statistical Genetics, lipids2013 |
| Obeng EA, Chappell RJ, Seiler M, Chen MC, Campagna DR, Schmidt PJ, Schneider RK, Lord AM, Wang L, Gambe RG, McConkey ME, Ali AM, Raza A, Yu L, Buonamici S, Smith PG, Mullally A, Wu CJ, Fleming MD, Ebert BL | 2016 | Effects of SF3B1 mutants on splicing in human erythropoiesis | http://www.ncbi.nlm.nih.gov/geo/query/acc.cgi?acc=GSE85712 | NCBI Gene Expression Omnibus, GSE85712 |

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
