## [Decision Letter]

Thank you for submitting your article "Gene-centric functional dissection of human genetic variation uncovers regulators of hematopoiesis" for consideration by *eLife*. Your article has been reviewed by three peer reviewers, including Stephen Parker as the Reviewing Editor, and the evaluation has been overseen by Mark McCarthy as the Senior Editor. The reviewers have opted to remain anonymous.

The reviewers have discussed the reviews with one another and the Reviewing Editor has drafted this decision to help you prepare a revised submission.

Summary:

Nandakumar et al. set out to study RBC GWAS signals in a model system of HSPCs differentiated to the erythroid lineage. They select ~400 genes across 75 loci and perform a shRNA screen quantifying shRNA abundance across differentiation time points. Their screen identifies 77 gene hits at 38 loci and they follow-up with some biological insight into two of these genes (TFR2 and SF3A2). Overall, this innovative approach provides a nice model for the field but we are unsure about a few important items that we elaborate on below.

Essential revisions:

1) The idea used to set up the screen and motivate gene selection by looking in the LD/recombination block needs to be further justified. Given most GWAS hits are non-coding, what is the any strong reason to assume the block must extend to the target gene, and how do we know most GWAS are not like the FTO locus that interacts with far-away genes? This is something that should be justified early on. As-is, this idea is only mentioned briefly in the Discussion.

2) As a corollary to point 1, a more conventional approach to connect GWAS to target genes is through eQTL analyses. Did the authors look at eQTL in whole blood (for which there are large data sets available) or other relevant cell types? How concordant are eQTL nominated genes with the genes revealed in the shRNA screen? The status of overlap and missingness would both be interesting to comment on here. Further, where eQTL data can nominate a target gene and the gene emerges from the shRNA screen, does the eQTL allelic direction of the RBC trait lead to reduced target gene expression (which would be consistent with the knock-down effects of shRNA treatment)? If the allelic direction of effect is different, how does one interpret this?

3) The screen quantified shRNA abundance as a function of erythroid differentiation and they identified targets that were either enriched or depleted late in the differentiation process. It is unclear what the significance of being enriched or depleted in the library means in terms of erythroid differentiation. Are there known genes that have biological functions which are expected to be enriched and depleted if they are depleted by shRNA?

4) Why do the authors use shRNA instead of CRISPR knockout or knockdown? There are numerous references (going back to early 2000s) documenting off-target effects of shRNA, so it is puzzling why this choice was made. For example, this recent preprint from George Daley's lab: "Short hairpin RNAs artifactually impair cell growth and suppress clustered microRNA expression." https://www.biorxiv.org/content/early/2018/12/11/372920 Given the "artifactual impair[ment] of cell growth" with shRNA, do the authors feel that their primary readout (growth) can be trusted given the shRNA perturbation?

5) Related to the concern mentioned in point 5, the authors should consider using CRISPRi (or CRISPR) to validate a subset of loci. Perhaps the two genes they've already focused on at the end. If such data are already (nearly) available (maybe in anticipation of such a reviewer comment), we encourage the authors to include it in this work. Although we list this in the essential revisions section and feel strongly about it, we realize this work may not be feasible within two months. If that's the case, we expect strong justification and text that directly addresses the point of shRNA vs CRISPR(i) screening approaches.

---

## [Author Response]

Essential revisions:1) The idea used to set up the screen and motivate gene selection by looking in the LD/recombination block needs to be further justified. Given most GWAS hits are non-coding, what is the any strong reason to assume the block must extend to the target gene, and how do we know most GWAS are not like the FTO locus that interacts with far-away genes? This is something that should be justified early on. As-is, this idea is only mentioned briefly in the Discussion.

Thank you for identifying this as an area for further clarification. While the method we use to nominate genes included in the shRNA screen may miss longer range interactions, when the screen was designed it seemed to represent a reasonable approach to capture a portion of the genetic targets for non-coding genetic variants and was similar to other approaches used to nominate genes that had been described in the literature. Since the screen was designed and executed, studies like Whalen and Pollard, 2019 have emerged which provide an enhanced sense of which topological tools at our disposal can offer the most comprehensive approach to target selection. While the use of strong LD partners by itself may not capture a majority of gene targets, by supplementing it with extension to the nearest recombination junctions and by adding wingspans on to the 5’ and 3’ end of candidate genes, we do actually enter into a regime where we would estimate we could resolve over half of likely targets. In Whalen and Pollard, 2019, the median eQTL distance is noted to be 49 kb, which falls well within the bulk of our probability density for the size of the SNP windows we are generating. Regardless, we agree that this issue could have been explored more deeply in the text, so we have made amendments to reflect that discussion.

Results:

“We endeavored to select candidate genes that could potentially underlie these 75 GWAS signals. […] However there still exists a nontrivial amount of valid targets within reach of proximity LD approaches, especially when the calculation of such windows are extended to reach the nearest recombination hotspot, suggesting that our approach would capture many candidate target genes.”

2) As a corollary to point 1, a more conventional approach to connect GWAS to target genes is through eQTL analyses. Did the authors look at eQTL in whole blood (for which there are large data sets available) or other relevant cell types? How concordant are eQTL nominated genes with the genes revealed in the shRNA screen? The status of overlap and missingness would both be interesting to comment on here. Further, where eQTL data can nominate a target gene and the gene emerges from the shRNA screen, does the eQTL allelic direction of the RBC trait lead to reduced target gene expression (which would be consistent with the knock-down effects of shRNA treatment)? If the allelic direction of effect is different, how does one interpret this?

To examine the eQTL-based approach and how it might relate to our results, we took the whole blood gene expression summary statistics from GTEx and intersected them with genomic regions in linkage disequilibrium r^2^ > 0.8 with the 75 sentinel SNPs from van der Harst et al. study used to identify the library of genes targeted in the screen (see Figure 1A). These regions were padded to a fixed 100 kb, as many of the regions are fairly small. This yielded 139 genes that one could argue would be nominated on an eQTL-basis from the total pool of 8661 examined. We performed a Fisher’s exact test on the contingency table comparing hits from our method of nomination with the set of eQTLs and eQTL-nominated genes. There were 35 hits (of the 77 total in our screen) present among the whole eQTL set, but 0 hits found among the 139 genes nominated by using eQTLs from whole blood. Thus, there was no appreciable connection between the eQTL-based method and our nomination method. We suggest that this observation also emphasizes how studies of variation in developing hematopoietic cells may not be accurately reflected by studies of circulating blood cells. We are looking at erythroblasts which are essentially not present in whole blood, so one would expect to miss cell type specific effects or eQTLs that act in early progenitor populations. Substantial additional work would be required to disentangle this further, but this is in keeping with our position that our nomination approach offers valuable information that is complementary to other typical means of nomination.

3) The screen quantified shRNA abundance as a function of erythroid differentiation and they identified targets that were either enriched or depleted late in the differentiation process. It is unclear what the significance of being enriched or depleted in the library means in terms of erythroid differentiation. Are there known genes that have biological functions which are expected to be enriched and depleted if they are depleted by shRNA?

We apologize if we did not do an adequate job elaborating on this in the text and connecting to the relevant panels (especially Figure 2C-E). We have made a number of changes which we hope address this point by more explicitly highlighting the outcomes of the included controls.

Results:

“Mobilized peripheral blood-derived primary human CD34^+^ HSPCs from 3 independent healthy donors were infected with a lentiviral-based pooled shRNA library consisting of 2803 hairpins targeting the 389 GWAS-nominated genes, along with 30 control genes (Moffat et al., 2006). […] The set of control shRNAs encompassed essential housekeeping genes as positive controls, negative controls in the form of luciferase and other genes not expressed in humans, and a well-established set of genes known to be important for erythropoiesis (erythroid controls) (Figure 1B).”

“The infected HSPCs were cultured using a three-phase semi-synchronous erythroid differentiation method where differentiation blockade reduces cell numbers either through cell death or through a failure of proliferation (Giani et al., 2016; Hu et al., 2013). We hypothesized that hairpins targeting potential regulators of erythropoiesis would be depleted or enriched during the three-phase erythroid culture, similar to our prior experience in analyzing specific GWAS-nominated genes (Giani et al., 2016; Sankaran et al., 2012; Ulirsch et al., 2016).”

“The tested set of hairpins targeting genes nominated by the 75 loci showed a variety of activities, forming a broad distribution spanning both decreases and increases in abundance at different time points (Figure 2B). […] Hairpins targeting genes with known biological roles in erythropoiesis, such as GATA1 and RPS19 (Khajuria et al., 2018; Ludwig et al., 2014), showed markedly decreased abundance across the time course.”

4) Why do the authors use shRNA instead of CRISPR knockout or knockdown? There are numerous references (going back to early 2000s) documenting off-target effects of shRNA, so it is puzzling why this choice was made. For example, this recent preprint from George Daley's lab: "Short hairpin RNAs artifactually impair cell growth and suppress clustered microRNA expression." https://www.biorxiv.org/content/early/2018/12/11/372920 Given the "artifactual impair[ment] of cell growth" with shRNA, do the authors feel that their primary readout (growth) can be trusted given the shRNA perturbation?

We understand the reviewers’ concern that shRNAs can have off-target effects and can impair cell growth. CRISPR based screening approaches can potentially overcome these issues. We chose to perform our screen in primary human CD34^+^ hematopoietic stem and progenitor cells (HSPCs), instead of using cell lines, to identify the bonafide regulators of erythropoiesis that could underlie common genetic variation. We have tested both CRISPR and shRNA-based approaches in primary HSPCs and our data, which we now added to Figure 1—figure supplement 2, suggest that CRISPR based pooled screens are not feasible to perform in primary HSPCs, given low levels of infection that are achieved. On the other hand, in our experience, results from shRNA experiments can be independently validated using CRISPR based approaches (Gianni et al., 2016). These results have motivated us to perform an shRNA screen in primary HSPCs and we can increase our confidence in the growth readout by employing 5-7 shRNAs per gene and compared them against a pool of negative control shRNAs.

We observe that all in one CRISPR/ CRISPRi lentiviral constructs are produced at lower viral titer (~5e5 infectious units /ml unconcentrated) compared to shRNA constructs (~ 9e6 infectious units /ml unconcentrated). This is likely due, at least in part, to reduced packaging ability, owing to the large size of the Cas9 cDNA (~4.2kb) compared to shRNAs (21bp) (Figure 1—figure supplement 2A). This low viral titer becomes a bottleneck when performing CRISPR screens in primary CD34^+^ HSPCs that require high multiplicities of infection (MOI) for successful gene transfer compared to rapidly dividing cell lines that require a low MOI.

In our initial studies, we observed that shRNA constructs were able to achieve good infectivity in CD34^+^ HSPCs at low MOI (40-60% infectivity at MOI ~20) (Figure 1—figure supplement 2B). On the other hand, CRISPR/CRISPRi constructs require very high MOI (20- 30% infectivity at MOI 150) to achieve reasonable infectivity. We also observe that CRISPRi based approaches gave uniform knockdown compared to cutting with CRIPSR/Cas9 when we targeted a surface antigen, Duffy/DARC (Figure 1—figure supplement 2C). Based on our calculations we would require 3000 ml of virus to perform just one replicate (10 e6 CD34^+^ cells) of a CRISPR/CRISPRi based screen with sufficient library representation, which would certainly stretch and likely exceed what we are capable of achieving reasonably for such a functional screen.

A recent study published a few months ago has overcome this bottleneck by electroporation of Cas9 as a protein followed by infection with a pooled sgRNA lentiviral library, which can be produced at higher viral titers (Ting et al., 2019). This could be a promising means to use CRISPR in future screening approaches, although this approach is not available in the CRISPRi format.

Results:

“Since the majority of common genetic variation underlying RBC traits appears to act in a cell-intrinsic manner within the erythroid lineage, we decided to perturb the candidate genes during the process of human erythropoiesis (Giani et al., 2016, Sankaran et al., 2012, Sankaran et al., 2008, Ulirsch et al., 2016). […] Furthermore, shRNAs can act rapidly to achieve gene knockdown and thereby avoid compensatory effects that can occur when complete CRISPR knockout is achieved (Rossi et al., 2015), better recapitulating the subtle changes in gene expression that are characteristic of common genetic variation.”

*5) Related to the concern mentioned in point 5, the authors should consider using CRISPRi (or CRISPR) to validate a subset of loci. Perhaps the two genes they've already focused on at the end. If such data are already (nearly) available (maybe in anticipation of such a reviewer comment), we encourage the authors to include it in this work. Although we list this in the essential revisions section and feel strongly about it, we realize this work may not be feasible within two months. If that's the case, we expect strong justification and text that directly addresses the point of shRNA vs CRISPR(i) screening approaches.*

We appreciate this question to validate the shRNA screen results using orthogonal approaches such as CRISPR or CRISPRi. To this end, we have attempted to knockout one of the hits from the screen (TFR2) in HSPCs using Cas9 ribonucleoprotein (RNP) nucleofection. We selected 3 guides targeting TFR2 with high on-target scores based on the metrics reported by Doench et al., Nature Biotechnology, 2016. While we did observe reasonable editing rates, these were not stable over the time course of the experiment for currently unclear reasons. Since the edits are not stable, the outcomes of these experiments are hard to decipher without extensive additional characterization.